# A Lens into Interpretable Transformer Mistakes via Semantic Dependency

**Ruo-Jing Dong** [1]   **Yu Yao** [1]   **Bo Han** [2]   **Tongliang Liu** [1]

## Abstract

*Semantic Dependency* refers to the relationship between words in a sentence where the meaning of one word depends on another, which is important for natural language understanding. In this paper, we investigate the role of semantic dependencies in answering questions for transformer models, which is achieved by analyzing how token values shift in response to changes in semantics. Through extensive experiments on models including the BERT series, GPT, and LLaMA, we uncover the following key findings: 1). Most tokens primarily retain their original semantic information even as they propagate through multiple layers. 2). Models can encode truthful semantic dependencies in tokens in the final layer. 3). Mistakes in model answers often stem from specific tokens encoded with incorrect semantic dependencies. Furthermore, we found that addressing the incorrectness by directly adjusting parameters is challenging because the same parameters can encode both correct and incorrect semantic dependencies depending on the context. Our findings provide insights into the causes of incorrect information generation in transformers and help the future development of robust and reliable models.

## 1. Introduction

Large Language Models (LLMs) based on the transformer architecture such as BERT (Devlin et al., 2018), GPT(Radford et al., 2019; Brown, 2020), and LLaMA (Touvron et al., 2023) have demonstrated remarkable capabilities across various natural language tasks. Alongside their benefits, LLMs pose significant risks and challenges (Weidinger et al., 2021). For example, LLMs may intensify biases (Navigli et al., 2023; Taori & Hashimoto, 2023), produce toxic content (Gehman et al., 2020; Ousidhoum et al., 2021),

generate false information (Lin et al., 2021), exhibit hallucinations (Ji et al., 2023), leak sensitive training data (Carlini et al., 2021), and even engage in deception (OpenAI, 2023; Scheurer et al., 2024). Addressing these issues has led to the development of evaluation methods for LLM performance (Liang et al., 2022) and strategies aimed at mitigating harmful outputs (Ganguli et al., 2022; Bai et al., 2022).

Existing research has elucidated several reasons for the mistakes observed in LLMs. Studies have suggested that non-linearity, insufficient model averaging, and inadequate regularization lead to mistakes when encountering crafted adversarial examples (Chakraborty et al., 2018; Zhang et al., 2020). Additionally, the programmatic behavior of LLMs may lead to vulnerabilities under security attacks and produce harmful content (Kang et al., 2024). Competing objectives and mismatched generalization may cause the susceptibility of safety-trained LLMs (Wei et al., 2024). Studies also indicate various reasons for language models generating unfaithful or nonsensical text, including source-reference divergence in data, imperfect representation learning, erroneous decoding, exposure bias, and parametric knowledge bias (Ji et al., 2023). These studies have identified various reasons that lead to mistakes and enhanced our understanding, providing valuable insights into model weaknesses. Building upon these insights, we aim to further explore the internal mechanisms within the model's architecture that lead to mistakes.

We believe that mistakes produced by LLMs arise from the way semantic dependencies, the relationships where the meaning of one token depends on another (Mel'čuk, 2001), are encoded within transformer models. Intuitively, in transformer models, inputs are tokenized and embedded as vectors that carry semantic information. These tokens then pass through multiple attention layers, where they exchange and aggregate semantic information to build semantic dependencies. These dependencies are crucial for the model's contextual understanding and reasoning, enabling it to generate coherent outputs. Since the model's predictions ultimately depend on the final layer outputs, which are constructed from the representations generated by earlier layers, any inaccuracies in the propagation and exchange of semantic information can result in misrepresented semantic dependencies. Such mistakes disrupt the model's ability to accurately understand token relationships and contextual

[1]Sydney AI Centre, The University of Sydney [2]TMLR Group, Hong Kong Baptist University. Correspondence to: Tongliang Liu <tongliang.liu@sydney.edu.au>.

meaning, thereby leading to mistakes in predictions.

To systematically explore the role of semantic dependencies in causing model mistakes, we propose a method to interpret the semantic information aggregation mechanisms of transformer models. The idea is that altering the semantic information of an input token should result in significant changes in the outputs of tokens that depend on this input information, while tokens that are not semantically dependent remain relatively unchanged. By evaluating the variations in output token representations in response to perturbations in input tokens, we can effectively trace the flow and aggregation of semantic information throughout the model. This approach allows us to gain insights into how semantic dependencies are encoded and propagated within transformer models and helps identify potential disruptions that may contribute to prediction mistakes.

**Key Findings.** In our exploration, we analyzed different transformer models such as BERT, LLaMA, and GPT. Here, we explain several key findings regarding the behavior of tokens for semantic information aggregation and propagation.

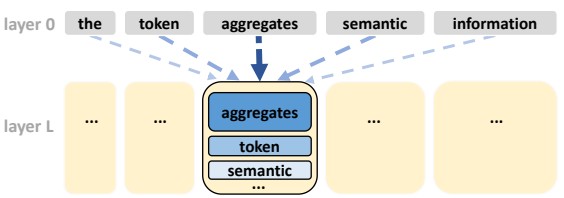

Figure 1: An Illustration of Finding 1.

1). **Most tokens primarily retain their original semantic information, even as they pass through the layers of transformers**. For example, in Figure 1, the arrows indicate the semantic information flow from the token at layer 0 to token at layer L. For the token "*aggregates*" in the input token sequence in layer 0, the final layer's token aggregates a large amount of information from its input token and a small amount of information from other tokens. The fact that most tokens still predominantly reflect their initial semantics highlights model's strong retention property, which is not inherently expected given the iterative aggregation of semantic information across many layers.

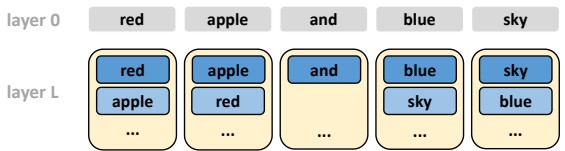

Figure 2: An Illustration of Finding 2.

2). **A token in the final layer usually encodes truthful *semantic dependency*.** Beyond preserving individual token semantics, the model must also encode truthful semantic

dependencies between tokens to understand sentence-level meaning. For example, the model must recognize how adjectives modify nouns or how subjects relate to actions. In the case of the input "red apple and blue sky" shown in Figure 2, an output token will encode the semantically **dependent** information "red" and "apple" together, rather than encoding semantically **independent** information like "blue" and "apple". We find our evaluated models can encode truthful semantic dependency in the final layer tokens.

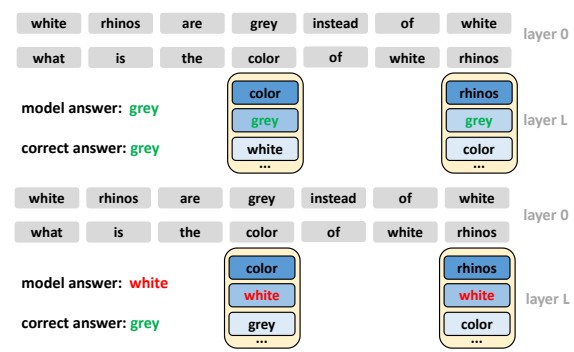

Figure 3: An Illustration of Finding 3.

3). Finally, we found that **when the model makes mistakes, certain tokens incorrectly encode information that is not semantically dependent.** For example, Figure 3 demonstrates that semantic information is aggregated differently in the output token sequence when the model outputs an incorrect answer. In a question-answering task where the context sequence "*white rhinos are grey instead of white*" is paired with the question "*What is the color of white rhinos?*", the correct answer is "*grey*". However, when the model incorrectly outputs "*white*", the question's key terms, such as "color" and "rhinos", contain more information about "*white*" rather than "*grey*". This highlights how false semantic dependency encoded in key tokens can lead to incorrect outputs.

4). Additionally, we also find that the encoded semantic dependencies within a token are highly sensitive to both **irrelevant** context changes and the order of contexts. Due to space limitations, a detailed introduction, experiments, and results are provided in Appendix A.2.

**Implications** Our insights into semantic dependency encoded within tokens of transformer models potentially help design new transformer architectures to be more resilient and semantically coherent. For example, our finding reveals that model mistakes often result from certain tokens erroneously encoding semantic dependencies that should not exist. To address this, future research could refine attention mechanisms to better prioritize meaningful token interactions and reduce the impact of the adversarial context. This could be achieved by implementing dynamic re-weighting

strategies in top QA task-specific attention heads we localized and incorporating stricter regularization techniques that can prevent tokens from erroneously encoding false dependencies while cautiously preserving the truthful ones.

Our paper is structured as follows: first, in Section 3, we verify a critical prerequisite for studying token-level semantic dependency: whether a final-layer token retains its original semantic information. Because to define a matched dependency between an input token (e.g., *red*) and another token (e.g., *apple*), the corresponding final-layer token *apple* or *red* must reflect its original meaning. The result confirms our method's validity to capture semantic dependencies across multiple layers. Next, in Section 4, we systematically verify if models can encode truthful semantic dependency. Our finding indicates while models generally possess this foundational capability, mistakes still occur. Therefore, in Section 5, we further investigate how mistakes arise from false semantic dependencies in QA tasks and develop a method to pinpoint a group of attention head parameters responsible for token-level semantic dependency. We testify that addressing the incorrectness by directly pruning parameters is challenging because the same parameters can encode both correct and incorrect semantic dependencies.

## 2. Related Work

**Semantic Dependency Parsing and Semantic Role Labeling.** Semantic dependency parsing (SDP) (Björkelund et al., 2010; Dozat & Manning, 2018) aims to identify semantic relationships between words in a sentence. Closely related to SDP, semantic role labeling (SRL) (He et al., 2017; Chen et al., 2025) focuses on identifying the predicate-argument structure of a sentence by assigning roles to words or phrases. However, the internal mechanisms by which transformer models encode, propagate, and utilize semantic dependencies remain largely opaque. Our work bridges this gap by exploring how internal mechanisms contribute to semantic dependency encoding and how these insights can be leveraged to address mistakes.

**Semantic Information Flow in Transformer.** Existing works (Liao et al., 2021; Schuster et al., 2022; Elhoushi et al., 2024) have studied activation stability and the limited contribution to token refinement in later layers of transformer models. However, whether the last-layer token retains its original semantic information in the input layer remains unexplored. Previous study (Geva et al., 2023) analyzes how factual associations are recalled while our study addresses a gap by studying how semantic dependencies are encoded in tokens and influence QA tasks.

**Interpretable Model Mistake Based on Attention Heads.** Previous studies highlight the roles of specific attention heads in model performance, such as retrieval heads for retrieving factual information (Wu et al., 2024) or property-specific attention heads in CLIP models (Gandelsman et al., 2024). Our work offers another perspective by interpreting model mistakes via token-level semantic dependency encoding, which provides insights into understanding and correcting model mistakes under specific question-answering cases. Additionally, our finding in mutual attention heads responsible for key dependencies in QA tasks also shows the importance of pruning parameters in attention heads without tempering the correct semantic dependency encoding.

**Probing Study for Linguistic Properties in Transformers.** Probing methods (Rogers et al., 2021) analyze the internal representations of pre-trained models to determine whether specific linguistic properties are encoded. For instance, research shows that BERT captures syntactic tree structures (Hewitt & Manning, 2019), semantic roles, and entity types (Tenney, 2019). Studies also quantify the mutual information between representations and linguistic properties (Pimentel et al., 2020). Token ablation is a parameter-free probing technique, which is widely used to analyze syntactic subtree structures (Wu et al., 2020), syntactic agreement (Finlayson et al., 2021), and bias (Vig et al., 2020). Naturally occurring morpho-syntactic perturbations is also used to probe dependencies (Amini et al., 2023). Unlike these studies, which focus on static linguistic features, we investigate token-level semantic dependency encoding, introducing a framework to quantify dependency strength without prior knowledge. Our approach captures context-sensitive semantic dependencies, which can vary across diverse scenarios.

**Feature Attribution and Binding Study.** Feature attribution methods primarily aim to assess the importance of individual tokens or features to the model's output. For example, prior work on attention flow (Abnar & Zuidema, 2020) quantifies token importance through accumulated attention matrices. Gradient-based techniques like Conservative Propagation (Ali et al., 2022) is used to assess token attribution. Our study shifts focus from token importance to the semantic dependencies encoded in token representations and how these affect model behavior.

Existing semantic dependency methods based on feature/token interactions (Eberle et al., 2020; Janizek et al., 2021; Schnake et al., 2021) mainly focus on studying the contribution of combinations of features or tokens to model predictions. Meanwhile, feature binding methods (Feng & Steinhardt, 2023; Vasileiou & Eberle, 2024; Wattenberg & Viégas, 2024) often do not test whether the model's most confident output reflects encoded semantic dependencies; rather, many assume this relationship holds and study downstream properties. In contrast, our method is designed to explicitly test the assumption by evaluating whether there is a dependence between the model's output and the semantic dependency encoded in the final-layer token.

# 3. Most Tokens Primarily Retain Their Original Semantic Information Through Transformer Layers

In this section, we introduce a perturbation-based method to explore some mechanisms of semantic information propagation in transformers. Extensive experiments show 1). even through multiple transformer layers, most final-layer tokens still primarily maintain their original semantic information from the first layer; 2). every final-layer token contains semantic information from almost all tokens (including itself) of the entire sequence. The results indicate our proposed method can effectively capture semantic changes in the input text and is suitable for detecting token-level semantic dependencies.

**Transformer Architecture**    We consider a general $L$-layer transformer model. Each layer consists of a multi-head self-attention mechanism (MHA) followed by a position-wise feed-forward network (FFN), along with residual connections. The input sequence of $N$ tokens is embedded into $D$-dimensional vectors and combined with positional encodings to form the initial representations:

$$\mathbf{z}^0 = [\mathbf{z}_1^0, \mathbf{z}_2^0, \dots, \mathbf{z}_N^0], \tag{1}$$

where $\mathbf{z}_i^0 \in \mathbb{R}^D$ is the embedding of the $i$-th token in 0-th layer.

In transformer-based models, the token sequence is updated through $L$ layers using the following two steps, where multi-head attention (MHA) and feed-forward networks (FFN) work together to enrich the text representations:

$$\hat{\mathbf{z}}^l = \text{MHA}^l(\mathbf{z}^{l-1}) + \mathbf{z}^{l-1}, \quad \mathbf{z}^l = \text{FFN}^l(\hat{\mathbf{z}}^l) + \hat{\mathbf{z}}^l, \tag{2}$$

where $l = 1, 2, \dots, L$. Here, $\text{MHA}^l$ and $\text{FFN}^l$ denote the multi-head attention and feed-forward network operations at layer $l$, respectively. The residual connections ensure that information flows directly through layers, facilitating the retention of original semantic information. For the $i$-th token in the output of the $L$-th layer, we have:

$$\mathbf{z}_i^L = \mathbf{z}_i^0 + \sum_{l=1}^{L} \text{MHA}_i^l(\mathbf{z}^{l-1}) + \sum_{l=1}^{L} \text{FFN}_i^l(\hat{\mathbf{z}}^l), \tag{3}$$

where $\text{MHA}_i^l$ and $\text{FFN}_i^l$ represent the operations affecting the $i$-th token at layer $l$ (Vaswani et al., 2017). Here we use the formulation proposed in the study (Gandelsman et al., 2024), which ignores the layer-normalization term. The above equation shows that a last-layer token can be written as a combination of first-layer tokens. This suggests that a last-layer token incorporates a varying yet unquantified amount of semantic information derived from the entire token sequence. Our research aims to address this gap by measuring such token-level semantic contribution.

Based on the above equation, we identify two underexplored mechanisms for validation. 1). **Self-information retention**: validate whether the $i$-th token $\mathbf{z}_i^L$ in the output layer primarily retains information about the $i$-th token $\mathbf{z}_i^0$ in the input layer. Specifically, we compare the changes of all tokens in the final layer $L$ with the changes in $\mathbf{z}_i^0$. If $\mathbf{z}_i^L$ changes most significantly when $\mathbf{z}_i^0$ changes, it suggests the $i$-th token in the final layer contains most information derived from the $i$-th token in the first layer. 2). **Sequence-level semantic aggregation**: validate whether a token in $L$-th layer aggregates semantic information from tokens of the entire sequence $z^0$. If every token change in $z^0$ leads to the change of $\mathbf{z}_i^L$, it suggests $\mathbf{z}_i^L$ contains information from all tokens.

**Token Perturbation**    We then generate $K$ perturbed versions of the input token $\mathbf{z}^{0(\text{org})}$ by only replacing the $i$-th token $\mathbf{z}_i^0$ with randomly sampled tokens from the vocabulary $\mathcal{V}$. Specifically, we sample a new token $\tilde{\mathbf{z}}_i^{0(k)}$ for $k$ times as follows.

$$\text{original } \mathbf{z}^{0(\text{org})} = [\mathbf{z}_1^0, \dots, \mathbf{z}_i^0, \dots, \mathbf{z}_N^0];$$

$$\text{perturbed } \tilde{\mathbf{z}}^{0(k)} = [\mathbf{z}_1^0, \dots, \tilde{\mathbf{z}}_i^{0(k)}, \dots, \mathbf{z}_N^0], \tag{4}$$

$$\text{where } \tilde{\mathbf{z}}_i^{0(k)} \sim \text{Uniform}(\mathcal{V}) \quad \text{and } k \in \{1, \dots, K\}.$$

Each perturbed sequence of token $\tilde{\mathbf{z}}^{0(k)}$ is processed independently through the $L$-layer transformer model, yielding $L$-layer token $\tilde{\mathbf{z}}^{L(k)}$. Similarly, the corresponding $L$-layer token for $\mathbf{z}^{0(\text{org})}$ is $\mathbf{z}^{L(\text{org})}$.

**Measuring Semantic Dependency**    To quantify how the perturbation of the $i$-th token $\mathbf{z}_i^0$ in the first layer affects $j$-th token $\mathbf{z}_j^0$ in final layer, we examine the average change of the $j$-th token across the $K$ sequences. Specifically, for the $j$-th token, we calculate the semantic dependency score $\Delta_{\mathbf{z}_j^L | \mathbf{z}_i^0}$, which is achieved by calculating average change $\Delta_{\mathbf{z}_j^L | \mathbf{z}_i^0}$ between its value in the original sequence and its values in the perturbed sequences:

$$\Delta_{\mathbf{z}_j^L | \mathbf{z}_i^0} = \frac{1}{K} \sum_{k=1}^{K} \left\| \tilde{\mathbf{z}}_j^{L(k)} - \mathbf{z}_j^{L(\text{org})} \right\|_2. \tag{5}$$

A higher value of $\Delta_{\mathbf{z}_j^L | \mathbf{z}_i^0}$ indicates that the $j$-th token in final layer $L$ is more sensitive to change of the $i$-th token. It implies that $j$-th token encodes more information from the $i$-th token, i.e. encodes a stronger semantic dependency.

To validate that the $j$-th token $\mathbf{z}_j^L$ in the output layer $L$ encode strongest semantic dependency with the $i$-th token in the input layer $\mathbf{z}_i^0$, we compare the average change $\Delta_{\mathbf{z}_j^L | \mathbf{z}_i^0}$ for all tokens. If $\Delta_{\mathbf{z}_i^L | \mathbf{z}_i^0}$ is the largest among all $\Delta_{\mathbf{z}_j^L | \mathbf{z}_i^0}, j \in \{1, \dots, N\}$, we can determine the $j$-th token in the final layer encode strongest semantic dependency with $i$-th token. For the first validation across multiple instances,

Table 1: Two validations on basic mechanisms of token-level semantic information propagation. Validation 1 is for self-information retention. Validation 2 is for sequence-level semantic aggregation.

| Model | validation 1 (%) | validation 2 (%) |
|---|---|---|
| BERT (encoder only) | 98.81 | 99.29 |
| RoBERTa (encoder only) | 93.06 | 94.69 |
| TinyRoBERTa (encoder only) | 94.29 | 96.40 |
| ALBERT (encoder only) | 97.01 | 97.74 |
| DistilBERT (encoder only) | 95.11 | 96.06 |
| DeBERTa (encoder only) | 99.62 | 99.74 |
| MobileBERT (encoder only) | 96.49 | 99.37 |
| MiniLM (encoder only) | 88.69 | 93.42 |
| GPT-2 (decoder-only, auto-regressive) | 75.15 | 100.00 |
| LLaMA3 (decoder-only, auto-regressive) | 95.59 | 100.00 |

we calculate the percentage $P$ that the $i$-th token's perturbation in input layer primarily affects its corresponding output token $\mathbf{z}_i^L$ in a transformer-based model $f_\theta$ on $M$ tested token cases as follows:

$$P(f_\theta) = \frac{1}{M} \sum_{m=1}^{M} \mathbb{1}_{\{i = \arg\max_{j=1}^{N} \Delta_{\mathbf{z}_j | \mathbf{z}_i^0}\}}. \qquad (6)$$

For the second validation, we calculate the percentage of cases when each input token $\mathbf{z}_i^0$ affect each output token $\mathbf{z}_j^L$, i.e., $\Delta_{\mathbf{z}_j^L | \mathbf{z}_i^0} > 0$, for all $i, j \in \{1, \ldots, N\}$

**Experiments** We validate model's self-information retention and sequence-level semantic aggregation using various sentences from six datasets, including gsm8k (Cobbe et al., 2021), Yelp (Zhang et al., 2015), GLUE (Wang et al., 2019), CNN/DailyMail (Hermann et al., 2015), OpenOrca (Lian et al., 2023) and WikiText (Merity et al., 2016). For each model, over 100,000 token cases were evaluated for each datasets (each token perturbation is treated as one case, 600,000 cases in total). Our analysis involves 10 various Transformer-based models, including BERT (Devlin et al., 2018), RoBERTa (Liu, 2019), ALBERT (Lan, 2019), DistilBERT(Sanh, 2019), DeBERTa (He et al., 2020), MobileBERT (Sun et al., 2020), MiniLM (Wang et al., 2020), GPT (Radford et al., 2019), and LLaMA (Touvron et al., 2023). Noted that we compute changes for nearly all tokens (over 95%) in each sequence, excluding special tokens such as [CLS] and [SEP], which ensures a comprehensive assessment of the semantic dependency across the input.

**Results** The results in Table 1 summarize two key metrics: the first column represents the percentage of tokens that primarily retain their original semantic information, while the second column indicates the percentage of input tokens that propagate semantic information to other tokens. Compared to BERT and LLaMA, there is a part of tokens that do not preliminarily retain their original information in GPT. From this experiment, we can observe that most tokens primarily retain their original semantic information, even as they pass through the transformer layers. Additionally, we verify that

almost every final-layer token receives semantic information from every token (including itself) in the sequence.

## 4. A Final-layer Token Encodes Truthful Semantic Dependency

In the previous section, we observed that most tokens primarily retain their original semantic information. However, we also found that tokens not only retain their own semantic information but also integrate semantic information from all other tokens. In this section, we aim to verify whether a token usually contains semantically dependent information, i.e, encodes truthful semantic dependencies in the final layer. **Specifically, our method investigates if tokens encode more semantic information from semantically related words compared to unrelated words in the sequence.** We find that this holds for most tokens.

To assess whether a token effectively encodes truthful semantic dependency, we first randomly select a word $\mathbf{w}_i^0$. We then identify a group $G_{\mathbf{z}_i^0}$ containing the indices of semantically dependent tokens by leveraging semantic dependency parsing tools SpaCy (Honnibal et al., 2020), which parse the words in the sentence that are semantically dependent with $\mathbf{w}_i^0$, including both head and children in parsing tree and the word itself. SpaCy leverages a pre-trained neural network model to predict syntactic relationships between words, offering more comprehensive annotations than human labeling. Next, we estimate a semantically dependent token group $\hat{G}_{\mathbf{z}_i^0}$ by changing $\mathbf{z}_i^0$ and obtain the indices of top $K_{\text{top}}$ tokens most sensitive to the change of $\mathbf{z}_i^0$. Finally, we evaluate the alignment between $G_{\mathbf{z}_i^0}$ and $\hat{G}_{\mathbf{z}_i^0}$.

**Semantically Dependent Token Groups** A group $G_{\mathbf{z}_i^0}$ contains the indices of tokens semantically dependent on $\mathbf{z}_i^0$. To identify a semantically dependent token group $G_{\mathbf{z}_i^0}$, we leverage existing semantic dependency parsing methods to obtain the semantically dependent word group $W_{\mathbf{w}_i^0}$ of the word $\mathbf{w}_i^0$, then convert it into a token group [1]. Intuitively, dependency parsing analyzes the grammatical structure of a sentence, establishing relationships between *"head"* words and the words that modify them. For example, in the sentence *"The quick brown fox jumps over the lazy dog."*, the word *"fox"* is semantically related to word *"quick"*, *"brown"* and *"jumps"* based on their grammatical dependencies. Once $W_{\mathbf{w}_i^0}$ is identified, each word $\mathbf{w}_j$ in $W_{\mathbf{w}_i^0}$ is converted into its corresponding token indices, and $\mathbf{w}_i^0$ is converted into $\mathbf{z}_i^0$, forming $G_{\mathbf{z}_i^0}$.

**Estimated Semantically Dependent Token Group by Leveraging Token Perturbation** To estimate the semantically dependent token group $\hat{G}_{\mathbf{z}_i^0}$ for each token $\mathbf{z}_i^0$, we

---

[1] In our experiments, we do not consider the case when $\mathbf{w}_i^0$ is converted to subword tokens.

Table 2: Alignment scores that indicate how well individual tokens encode truthful semantic dependencies (%).

| Model | Average Alignment Score (%) |
|-------|------------------------------|
| BERT | 87.86 |
| RoBERTa | 87.71 |
| TinyRoBERTa | 82.44 |
| ALBERT | 88.77 |
| DistilBERT | 88.88 |
| DeBERTa | 87.17 |
| MobileBERT | 85.80 |
| MiniLM | 84.62 |
| GPT-2 | 93.41 |
| LLaMA3 | 92.47 |

measure semantic dependency score $\Delta_{\mathbf{z}_j^L | \mathbf{z}_i^0}$ by Eq. (5) for each token $\mathbf{z}_j^L$ in the final layer $L$. Then we rank it and select the largest $K_{\text{top}}$ indices within the sequence into a set denoted as $\hat{G}_{\mathbf{z}_i^0}$.

$$\hat{G}_{\mathbf{z}_i^0} = \{j \mid j \in \text{indices of } \max_{K_{\text{top}}}(\Delta_{\mathbf{z}_j^L | \mathbf{z}_i^0}, j = 1, \ldots, N)\}. \quad (7)$$

**Calculating Alignment Score**  To assess the alignment between our estimated semantically dependent token group $\hat{G}_{\mathbf{z}_i^0}$ and the semantically related token group $G_{\mathbf{z}_i^0}$, we compute the alignment score $S_i$ to measure the overlap between $\hat{G}_{\mathbf{z}_i^0}$ and $G_{\mathbf{z}_i^0}$:

$$S_{\mathbf{z}_i^0} = \frac{\left| G_{\mathbf{z}_i^0} \cap \hat{G}_{\mathbf{z}_i^0} \right|}{K_{\text{top}}}, \quad (8)$$

where $\left| G_{\mathbf{z}_i^0} \cap \hat{G}_{\mathbf{z}_i^0} \right|$ represents the number of overlapping tokens between $G_{\mathbf{z}_i^0}$ and $\hat{G}_{\mathbf{z}_i^0}$. A high alignment score means the tokens influenced by the perturbation of $\mathbf{z}_i^0$ tend to be the ones that are semantically related to it, indicating models' ability to encode truthful semantic dependency.

**Experiments and Results**  We conducted this experiment on 10 transformer models. We first construct a specialized word dependency dataset using SpaCy. This dataset includes sentences from the GLUE dataset, where each word (as one case) in the sentence is annotated with its semantically dependent word groups as standard dependency data. For each model, we evaluated over 10,000 cases, where each case corresponds to perturbing a single token and computing the alignment score. The average alignment scores across all cases are presented in Table 2. The overall high alignment scores indicate these models can generally encode truthful semantic dependencies in final-layer tokens.

# 5. When the Model Makes Mistakes, It Falsely Aggregates Semantically Independent Information within a Token

Models rely on correctly encoding semantic dependencies to generate coherent and contextually appropriate outputs;

otherwise, the resulting content may be random or confusing. Although Section 4 showed that transformer models can encode truthful semantic dependencies in the final layer, they still produce incorrect outputs in certain contexts. This suggests that correctly encoding most semantic dependencies is insufficient to prevent mistakes. We hypothesize that such mistakes arise from the model's tendency to encode false semantic dependencies in tokens through transformer layers. Intuitively, in the final layer, token representations are transformed via a linear prediction layer to produce output logits. However, the limited discriminative power of this linear layer makes it susceptible to errors when tokens encode false semantic dependencies from other unrelated or misleading tokens, ultimately leading to incorrect predictions. To test our hypothesis, we conduct an empirical analysis using the question answering (QA) task, which is particularly suitable for evaluating the influence of token-level semantic dependencies because QA inherently requires the model to understand and associate tokens in a question with those in the context.

In this section, we firstly identify that model mistakes in QA tasks stem from incorrect semantic dependencies encoded in question tokens. **Specifically, our method examines whether the semantic dependency strength between wrong answer tokens and question tokens exceeds that between correct answer tokens and question tokens.** Furthermore, to localize model parameters that encode semantic dependency, we propose a method to pinpoint a group of attention head parameters responsible for token-level semantic dependency. We demonstrate that directly pruning the parameters to correct these mistakes is challenging, as the same parameters may encode both correct and incorrect semantic dependencies.

## 5.1. Evaluation of Correct & False Semantic Dependencies in QA Task

To test our hypothesis that model errors often result from falsely aggregated independent semantic information within tokens, we analyze QA pairs where the language model outputs either the correct answer extracted from the context or an incorrect one. We then compare the semantic dependencies between tokens in incorrect answers and question tokens against those in correct answers within a question-answering (QA) task.

Consider the QA example illustrated in Figure 4, where the context provides the correct answer "national anthem" and an misleading phrase "sign language." If the BERT model incorrectly outputs "sign language" instead of "national anthem", this presents an opportunity to examine the underlying semantic dependencies between context tokens and question tokens that led to the mistake.

Formally, let $Q = \{\mathbf{q}_i^0\}_{i=1}^{N_Q}$ represent the set of tokens in

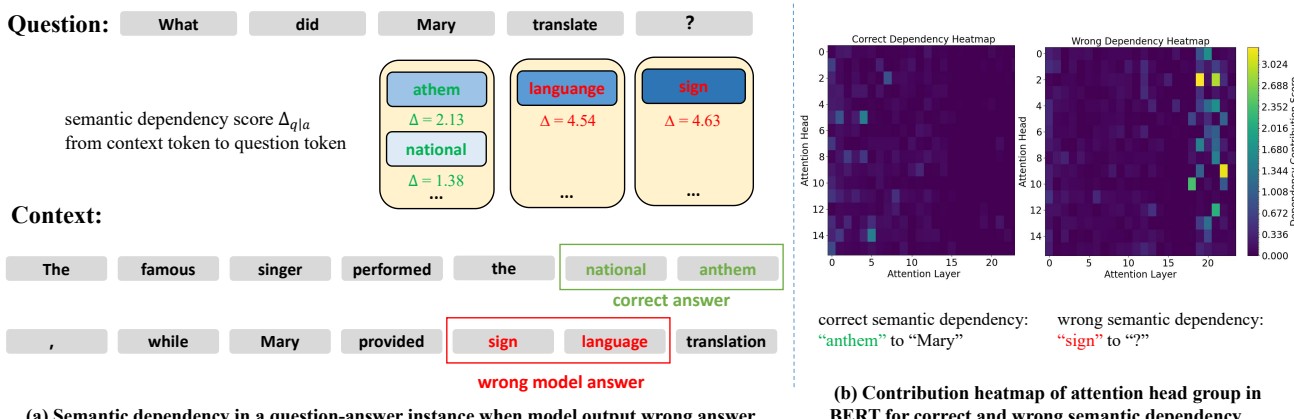

(a) Semantic dependency in a question-answer instance when model output wrong answer

(b) Contribution heatmap of attention head group in BERT for correct and wrong semantic dependency

Figure 4: A question-answer instance for false semantically dependent information within tokens.

the question, $A_{\text{correct}} = \{\mathbf{a}_i^0\}_{i=1}^{N_C}$ represent the correct answer tokens in the context, and $A_{\text{wrong}} = \{\mathbf{a}_i^0\}_{i=1}^{N_W}$ represent the incorrect answer tokens in the context. For each answer token $\mathbf{a}_i$, we measure its semantic dependency on each question token $\mathbf{q}_j \in Q$ by computing a semantic dependency score $\Delta_{\mathbf{q}_j^L|\mathbf{a}_i^0}$ by Eq. (5). This score quantifies the degree to which answer token $\mathbf{a}_i$ influences the question token $\mathbf{q}_j$ in the final layer $L$ of the model. Next, we determine the maximum semantic dependency score for each answer token by selecting the highest $\Delta_{\mathbf{q}_j^L|\mathbf{a}_i^0}$ across all question tokens

$$\Delta'_{\mathbf{a}_i^0|Q} = \max_{j=1}^{N_Q} \Delta_{\mathbf{q}_j^L|\mathbf{a}_i^0}.$$

For both correct and incorrect answers, we compute the highest dependency scores across all answer tokens:

$$\Delta'_{A_{\text{correct}}|Q} = \max_{k=1}^{N_C} \Delta'_{\mathbf{a}_k^0}, \quad \Delta'_{A_{\text{wrong}}|Q} = \max_{k=1}^{N_W} \Delta'_{\mathbf{a}_k^0}. \quad (9)$$

To evaluate whether the maximum dependency score for incorrect answers exceeds that of correct answers when a model makes mistakes, we calculate the percentage that $\Delta'_{A_{\text{wrong}}|Q}$ is greater than $\Delta'_{A_{\text{correct}}|Q}$ given the question and answer pairs where the model makes mistakes. Specifically,

$$P(f_\theta) = \frac{1}{H} \sum_{i=1}^{H} \mathbb{1}_{\{\Delta'_{A_{\text{wrong}}|Q} > \Delta'_{A_{\text{correct}}|Q}\}}, \quad (10)$$

where $H$ represents the total number of failed QA instances.

**Experiments**   We apply our evaluation method to the Stanford Question Answering Dataset (SQuAD) 1.1 (Rajpurkar et al., 2016), which comprises context paragraphs extracted from Wikipedia articles, along with manually crafted questions and their corresponding correct answers. Each QA instance in the dataset provides a context from which the correct answer is a continuous span of text, which means the answer exists verbatim in the context. Our analysis involves processing over 100,000 QA validation cases across 10 Transformer-based models.

Table 3: Two-by-two possibility table for model answer correctness and semantic dependency correctness.

|  | Correct Dependency | Incorrect Dependency |
|---|---|---|
| Answer Correctly | $P'(f_\theta)$ | $1 - P'(f_\theta)$ |
| Answer Incorrectly | $1 - P(f_\theta)$ | $P(f_\theta)$ |

For each QA instance, we first determine whether the model fails to output the correct answer by evaluating the F1 score between the model's predicted answer and the ground truth answer. We consider a prediction to be incorrect if the F1 score is below 0.6. Consequently, we collect these incorrect answer cases (where F1 < 0.6) for further analysis to examine the presence of false dependencies. This selection criterion ensures that we focus on substantial mistakes rather than minor discrepancies, thereby providing a robust basis for evaluating semantic dependency misalignments.

In these selected failed QA cases, we compare the semantic dependencies between question tokens and correct/incorrect answer tokens. For each case, we calculate whether the maximum semantic dependency score of incorrect answer tokens $\Delta'_{A_{\text{wrong}}|Q}$ exceeds that of correct answer tokens $\Delta'_{A_{\text{correct}}|Q}$. This comparison allows us to assess whether the model's mistakes are associated with false semantic dependencies from incorrect context tokens influencing question tokens. For successful QA cases, we calculate the percentage $P'(f_\theta)$ when the maximum semantic dependency score of correct answer tokens $\Delta'_{A_{\text{correct}}|Q}$ exceeds that of incorrect answer tokens $\Delta'_{A_{\text{wrong}}|Q}$, where the incorrect tokens are randomly sampled $T$ times from the remaining context tokens (excluding the correct answer tokens). Finally, we summarize $P(f_\theta)$ and $P'(f_\theta)$ for all models.

**Results**   For each model, we provide a two-by-two possibility table (shown in Table 3) for model answer correctness and semantic dependency correctness. $P(f_\theta)$ stands for the percentage when the model answers incorrectly and semantic dependency is incorrectly encoded. $P'(f_\theta)$ stands for

Table 4: Summarized percentages for two-by-two possibility table and F1 scores of all models. Results based on GPT evaluations are provided in the Appendix A.3.

|  | BERT | RoBERTa | tinyRoBERTa | ALBERT | DistilBERT | DeBERTa | MobileBERT | MiniLM | GPT-2 | LLaMA3 |
|---|---|---|---|---|---|---|---|---|---|---|
| $P(f_\theta)$ | 79.07 | 69.20 | 77.94 | 71.86 | 81.80 | 75.32 | 66.61 | 77.56 | 48.04 | 64.56 |
| $1 - P(f_\theta)$ | 20.93 | 30.80 | 22.06 | 28.14 | 18.20 | 24.68 | 33.39 | 22.44 | 51.90 | 35.44 |
| $P'(f_\theta)$ | 93.26 | 82.32 | 83.33 | 87.05 | 96.48 | 89.25 | 75.24 | 91.97 | 81.25 | 70.56 |
| $1 - P'(f_\theta)$ | 6.74 | 17.68 | 16.67 | 12.95 | 3.52 | 10.75 | 24.76 | 8.03 | 18.75 | 29.44 |
| Average F1 Score (%) | 92.93 | 84.86 | 82.83 | 80.56 | 85.71 | 91.69 | 81.19 | 85.34 | 0.78 | 35.81 |

the percentage when the model answers correctly and the semantic dependency is correctly encoded.

The final results for all QA cases and average F1 score are summarized in Table 4, which generally shows a significant proportion of model mistake cases across various models can be attributed to falsely encoded semantic dependencies. For instance, in BERT's case, the high percentage $P(f_\theta)$ implies that when the model selects an incorrect answer, it is more likely due to the erroneous answer tokens causing a stronger semantic influence on the question tokens than the correct answer tokens. Conversely, the overall high $P'(f_\theta)$ suggests when the model correctly encodes the semantic dependency in the final-layer token, it usually provides the correct answer. These findings highlight the importance of semantic dependency encoded in the final-layer token for model predictions.

The variation in Percentage across different models highlights inherent differences in how each architecture manages semantic dependencies and mitigates the impact of misleading information. For models like DistilBERT and BERT, the higher $P(f_\theta)$ suggests that their architecture may be more susceptible to false dependencies when mistakes occur. For models like RoBERTa and MobileBERT, the Lower $P(f_\theta)$ means false dependency accounts for a small proportion in failed QA instances, which means there may be some other factors that lead to wrong outputs. Unlike the BERT series, GPT-2 and LLaMA show much lower $P(f_\theta)$ and F1 scores. This discrepancy can be attributed to their generative nature where they tend to produce new words or synonyms rather than reproducing original ground-truth answers. Their logic of answer generation differs from that of the BERT-based QA models, potentially introducing inconsistencies in how semantic dependencies are encoded and impacting semantic dependency evaluation.

## 5.2. Localize Parameters of Attention Head Group Responsible for Semantic Dependency

As shown above, mistakes in QA tasks are closely tied to token-level semantic dependency. To better understand the network's role in model mistakes and evaluate whether false semantic dependencies can be adjusted to improve model performance, we focus on localizing the parameters that encode semantic dependencies. Our motivation stems from two key considerations: 1). If correct and incorrect dependencies are controlled by different parameters, we can directly modify the ones responsible for incorrect dependencies (e.g. pruning methods) to improve model performance. 2). Conversely, if they share the same parameters, we need other strategies to address the challenge of removing false dependencies without affecting the correct ones.

As discussed above, the attention mechanism plays a key role in propagating semantic information between tokens, ultimately enabling the final layer token to encode various semantic dependencies. Building on these foundations, we focus primarily on analyzing the contribution of attention head parameters in this paper. Specifically, we propose a method to identify attention heads primarily responsible for encoding specific token dependencies.

Inspired by previous study (Gandelsman et al., 2024), the contribution of $l$-th MHA on $j$-th token can be broken down into tokens and heads.

$$\text{MHA}_j^l(\mathbf{Z}^{l-1}) = \sum_{h=1}^{H} \sum_{i=1}^{N} x_i^{l,h}, \quad x_i^{l,h} = \alpha_i^{l,h} W_{VO}^{l,h} z_i^{l-1} \tag{11}$$

Specifically, for any token dependency, i.e., token dependency from $i$-th token to $j$-th token, including correct or wrong token dependency in QA tasks mentioned above, we replace the $i$-th the token with $K$ randomly sampled tokens. Then we measure each head's contribution to semantic dependency by calculating average change $\Delta_{\mathbf{z}_j^L|\mathbf{z}_i^0}^{l,h}$ between original head contribution and perturbed head contributions:

$$\Delta_{\mathbf{z}_j^L|\mathbf{z}_i^0}^{l,h} = \frac{1}{K} \sum_{k=1}^{K} \left\| x_i^{l,h(k)} - x_i^{l,h(\text{org})} \right\|_2 \tag{12}$$

As shown in Figure 4(b), we test the semantic dependency contribution score $\Delta_{\mathbf{q}_j^L|\mathbf{a}_i^0}^{l,h}$ of each attention head in BERT for both wrong semantic dependency between "sign" and "?" and correct semantic dependency between "anthem" to "Mary" in corresponding QA instance in Figure 4(a). The heatmap reveals the group of attention heads (highlighted in brighter colors) that mutually contribute to a semantic dependency in this specific context.

**Experiments**   To evaluate how parameters in attention heads contribute to semantic dependencies across various cases, we conducted experiments on different fine-tuned models commonly used in QA tasks with high accuracy (F1 > 0.8). Specifically, we analyzed the frequency of each top 5% contributing attention head for correct and false semantic dependency across all failed QA cases (shown in Figure 5). Due to space constraints, we present the results from the four highest-performing models and include additional experiments in Appendix A.4.

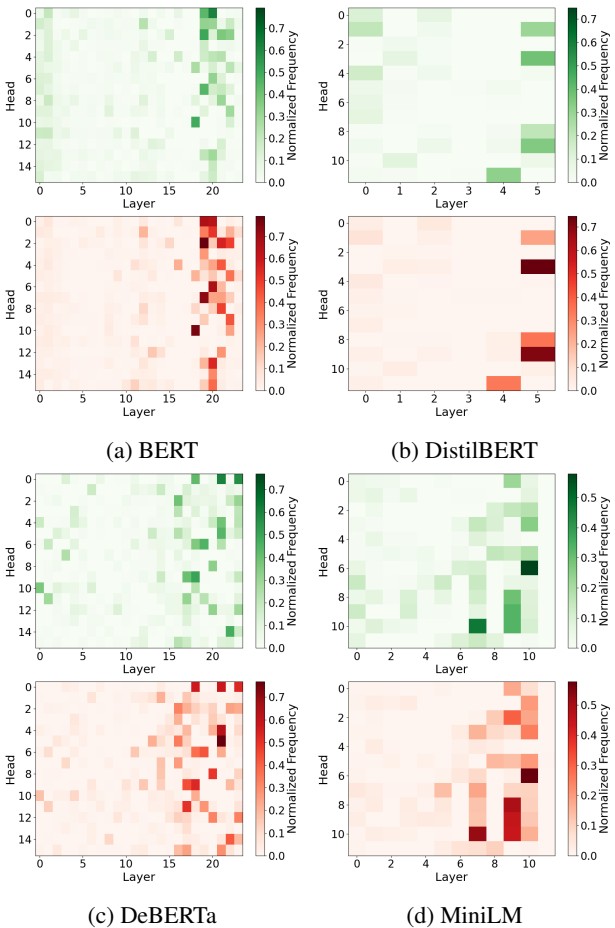

(a) BERT

(b) DistilBERT

(c) DeBERTa

(d) MiniLM

Figure 5: Frequency of each top 5% contributing attention head for correct (green) and false (red) semantic dependency across all failed QA cases.

**Results**   Figure 5 reveals a shared group of top attention heads responsible for both correct and false semantic dependency in these models. This suggests that model mistakes in QA tasks stem from the parameters of these specific attention heads. However, it also suggests false semantic dependencies cannot be reduced by directly disabling them (e.g., via head pruning approaches (Voita et al., 2019; Michel et al., 2019)) because these heads also contribute

to encoding correct semantic dependencies and indiscriminate removal of attention heads may inadvertently disrupt essential task-specific dependencies. Our finding highlights a critical technical challenge: how to disentangle and optimize attention mechanisms to suppress false dependencies while preserving correct ones. Future work may require more targeted re-weighting or regularization strategies to achieve this balance.

# 6. Discussion and Future Work

Our current method has certain limitations, which we believe present valuable opportunities for future work. Firstly, our analysis relies on perturbation-based approaches to assess token dependencies, which require that answer tokens appear in the context. This limits its applicability to scenarios where the model generates answers not directly found in the input. We aim to expand our ability to effectively analyze dependencies in such cases.

Additionally, perturbation involves removing existing information and introducing new information, which can cause variability in output tokens. For instance, replacing a token with a semantically similar yet different token may lead to significant variation depending on the model's interpretation. We address this by randomly sampling new tokens to ensure diversity and reduce bias, though some variability remains. Future work will focus on refining this calibration.

Our analysis primarily focuses on the semantic dependencies between final-layer tokens and first-layer tokens. This design choice is motivated by our goal of understanding errors in the model's output, where the final-layer token representations are expected to have the most direct influence, as supported by prior studies. While our current study centers on the final layer, our method is general and can be applied to intermediate layers as well. We will explore token dependencies across different layers in future work.

# 7. Conclusion

In this paper, we delved into the internal mechanisms of transformer models to explore how semantic dependencies are encoded in tokens, which can contribute to the mistakes produced by language models. Extensive experiments reveal that: 1) most tokens primarily retain their original semantic information across layers. 2) models can encode truthful semantic dependencies in final-layer tokens. and 3) model mistakes often stem from tokens encoding incorrect dependencies. However, shared attention head parameters help encode both correct and false dependencies, indicating the challenge of removing incorrect dependencies. We believe these insights can offer valuable implications for future transformer model design.

## Impact Statement

This paper presents work whose goal is to advance the field of Machine Learning. There are many potential societal consequences of our work, none which we feel must be specifically highlighted here.

## Acknowledgements

TL is partially supported by the following Australian Research Council projects: FT220100318, DP220102121, LP220100527, LP220200949, and IC190100031. BH was supported by RGC Young Collaborative Research Grant No. C2005-24Y and NSFC General Program No. 62376235.

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

# A. Appendix

## A.1. Detailed Related Works

**Semantic Dependency Parsing and Semantic Role Labeling.** Semantic dependency parsing (SDP) (Björkelund et al., 2010; Dozat & Manning, 2018) aims to identify semantic relationships between words in a sentence, by constructing a directed graph, where nodes represent words and edges capture their semantic dependencies. Closely related to SDP, semantic role labeling (SRL) (He et al., 2017; Chen et al., 2025) focuses on identifying the predicate-argument structure of a sentence by assigning roles to words or phrases based on their semantic relationship to a verb or predicate. Notably, studies (Li et al., 2004; Shen & Lapata, 2007; Khashabi et al., 2018) have shown that incorporating semantic role information enhances question-answering systems. However, despite their advancements, the internal mechanisms by which transformer models encode, propagate, and utilize semantic dependencies remain largely opaque. Our work bridges this gap by exploring how internal mechanisms contribute to semantic dependency encoding and how these insights can be leveraged to address mistakes.

**Semantic Information Flow in Transformer.** Existing work (Liao et al., 2021; Schuster et al., 2022; Elhoushi et al., 2024) have studied model activation stability in later layers of transformer models. Specifically, additional layers may contribute minimally to the refinement of token representations, which enables techniques like early exit to accelerate inference. However, whether the token in the last layer mostly contains its original semantic information in the input layer has not been studied. Previous study (Geva et al., 2023) analyzes how factual associations are recalled in auto-regressive language models, highlighting the roles of MLP sublayers in enriching subject representations and attention heads in extracting attributes. Our study addresses a gap by studying how semantic dependencies are encoded in tokens and influence QA tasks in both non-auto-regressive (BERT series) and auto-regressive models (GPT, LLaMA).

**Interpretable Model Mistake Based on Attention Heads.** Existing works have studied specific roles of attention heads to explain model mistakes. Study (Wu et al., 2024) identifies specific attention heads, termed retrieval heads, which are critical for retrieving factual information from long contexts. The absence or malfunctioning of these retrieval heads may lead to model mistakes. Another study (Gandelsman et al., 2024) shows some attention heads in CLIP have property-specific roles (e.g., location or shape), which are important for model performance. Our work offers another perspective by interpreting model mistakes via token-level semantic dependency encoding, which provides insights into understanding and correcting model mistakes under specific question-answering cases. Additionally, our finding in mutual attention heads responsible for key dependencies in QA tasks also shows the importance of adjusting parameters in attention heads without tempering the correct semantic dependency encoding.

**Probing Study for Linguistic Properties in Transformer.** Probing methods (Rogers et al., 2021) are widely used to analyze the internal representations of pre-trained language models to determine whether specific linguistic properties are encoded. A previous study demonstrated that BERT encodes syntactic tree structures in its vector space, allowing a probing classifier to reconstruct syntactic distances between words using linear transformations (Hewitt & Manning, 2019). Additionally, the study revealed that BERT encodes high-level linguistic features like entity types, semantic roles, and relations through probing tasks (Tenney, 2019). Moreover, existing research utilized information-theoretic probing methods to quantify the mutual information between model representations and linguistic properties, reducing over-interpretation risks (Pimentel et al., 2020).

Token ablation is a widely used, parameter-free probing technique. For example, researchers have studied the influence of syntactic subtree structures on masked language model (MLM) predictions through ablations (Wu et al., 2020). Others analyze syntactic agreement in language models through causal interventions, identifying key neurons and attention heads (Finlayson et al., 2021). Gender bias has been investigated using causal mediation analysis (Vig et al., 2020). Naturally occurring perturbations, which refer to sentences differing in specific morpho-syntactic features, have been used to probe causal relationships (Amini et al., 2023).

These works primarily investigate how models encode syntactic and high-level semantic features, such as entity relations or syntactic structures. In contrast, our study focuses specifically on token-level semantic dependency encoding, analyzing fine-grained interactions between individual tokens rather than task-specific feature aggregation or high-level semantic encoding. Moreover, we introduce an evaluation framework to measure semantic dependency strength between two tokens without relying on prior knowledge. Our approach also identifies false semantic dependencies that arise when the model

produces incorrect answers. Unlike static syntactic or semantic structures, our framework captures the dynamic and context-sensitive semantic dependencies, which can vary irregularly across diverse scenarios.

**Feature Attribution and Binding Study.** Feature attribution methods primarily aim to assess the importance of individual tokens or features to the model's output. For example, prior work on attention flow (Abnar & Zuidema, 2020) quantifies token importance through accumulated attention matrices. Gradient-based techniques like Conservative Propagation (Ali et al., 2022) is used to assess token attribution. Our study shifts focus from token importance to the semantic dependencies encoded in token representations and how these affect model behavior.

Existing semantic dependency methods based on feature/token interactions (Eberle et al., 2020; Janizek et al., 2021; Schnake et al., 2021) mainly focus on studying the contribution of combinations of features or tokens to model predictions. Meanwhile, feature binding methods (Feng & Steinhardt, 2023; Vasileiou & Eberle, 2024; Wattenberg & Viégas, 2024) often do not test whether the model's most confident output reflects encoded semantic dependencies; rather, many assume this relationship holds and study downstream properties. In contrast, our method is designed to explicitly test the assumption by evaluating whether there is a dependence between the model's output and the semantic dependency encoded in the final-layer token.

## A.2. Extra Finding: The Semantic Dependency Encoded in a Token Is Influenced by Both Irrelevant Context Changes and Order of Contexts

In this section, we study whether **the rank of semantic dependency strength encoded in a token changes when adding irrelevant context or simply changing the order of the context sequence**. We also found some interesting phenomena after experiments: **1). Semantically related tokens remain relatively stable when altering irrelevant context or the order of the context. 2). Left context change usually causes greater influence than right context change.**

Robustness studies have demonstrated that the inclusion of irrelevant context (Shi et al., 2023) or adversarial sentences (Jia & Liang, 2017) in prompts can lead to a significant decline in model accuracy. They usually work by analyzing model performance on various types of adversarial examples and attribute the decline to broader issues, such as the model's tendency to rely on surface-level features like word overlap and positional cues. To further explore the underlying reason for such performance decline from a token-level perspective, we test whether altering the irrelevant context or rearranging the order of independent sentences affects the rank of semantic dependency strength.

For example, we have two semantically independent token sequences "*white rhinos are gray*" and "*apples are red*" in Figure 6, where "apples are red" (highlighted with green background) serves as irrelevant context to "white rhinos are gray". On the top half part of the figure, when we add the irrelevant context "apples are red", the rank of semantic dependency strength between the token "rhinos" and tokens in its sequence "white rhinos are gray." varied. On the bottom part of the figure, the same thing happens when we maintain the overall input semantic information unchanged and only change the order of the two token sequences. This demonstrates that even when two token sequences are semantically independent, irrelevant changes in context and the ordering of sequences can significantly alter how semantic information is aggregated within each token.

**Semantic Dependency Analysis with Irrelevant Context Change** To validate whether irrelevant context influences the semantic dependencies of tokens in a sequence, we selected two semantically independent sentences randomly sampled from a dataset. Consider two sentences:

"*The sky is blue.*" vs "*The apple is red. The sky is blue.*", i.e., $s_1$ vs $(s_2, s_1)$

"*The sky is blue.*" vs "*The sky is blue. The apple is red.*", i.e., $s_1$ vs $(s_1, s_2)$

We investigated whether the semantic dependencies within "*The sky is blue.*" remain unchanged when appended with "The apple is red." on its left side or right side. Since both contexts are independent, with no semantic dependencies between them, the semantic dependencies within "*The sky is blue.*" should remain unchanged regardless of their surrounding context in the input sequence.

Specifically, given two input token sequences are $\mathbf{z}^{0(s_1)} = \{\mathbf{z}_i^0\}_{i=1}^{N_1}$ and $\mathbf{z}^{0(s_2)} = \{\mathbf{z}_j^0\}_{j=1}^{N_2}$, respectively. Here, we validate the semantic dependencies within $\mathbf{z}^{0(s_1)}$. We created two additional token sequences: $\mathbf{z}^{0(\text{Left})} = [\mathbf{z}^{0(s_2)}, \mathbf{z}^{0(s_1)}]$

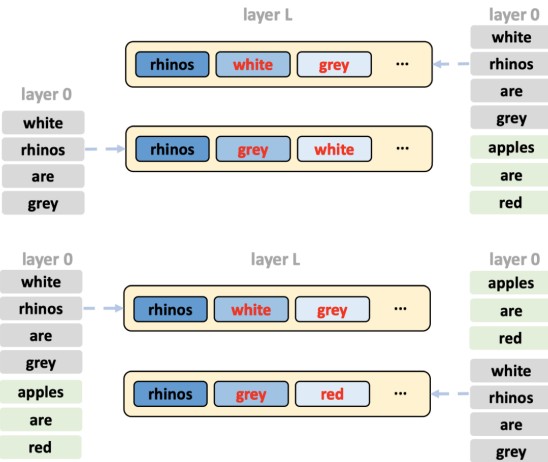

Figure 6: Semantic information propagation is influenced by irrelevant context change and sequence order change.

and $\mathbf{z}^{0(\text{Right})} = [\mathbf{z}^{0(s_1)}, \mathbf{z}^{0(s_2)}]$, where $\mathbf{z}^{0(\text{Left})}$ is obtained by concatenating $\mathbf{z}^{0(s_2)}$ to the left and $\mathbf{z}^{0(\text{Right})}$ is obtained by concatenating $\mathbf{z}^{0(s_2)}$ to the right. For token $\mathbf{z}_i^0$ from $\mathbf{z}^{0(s_1)}$, we obtain the corresponding estimated semantic dependency token group $\hat{G}_{\mathbf{z}_i^0}^{s_1}$ via Eq. (7). By using the same approach, estimated semantic dependency token groups $\hat{G}_{\mathbf{z}_i^0}^{\text{Left}}$ and $\hat{G}_{\mathbf{z}_i^0}^{\text{Right}}$ for $\mathbf{z}^{0(\text{Left})}$ and $\mathbf{z}^{0(\text{Right})}$ can also be obtained. Then the Dependency Alteration Score (DAS) of $\hat{G}_{\mathbf{z}_i^0}^{\text{Left}}$ and $\hat{G}_{\mathbf{z}_i^0}^{s_1}$ can be calculated as follows:

$$\text{DAS}(\hat{G}_{\mathbf{z}_i^0}^{\text{Left}}, \hat{G}_{\mathbf{z}_i^0}^{s_1}) = 1 - \frac{\text{LCS}(\hat{G}_{\mathbf{z}_i^0}^{\text{Left}}, \hat{G}_{\mathbf{z}_i^0}^{s_1})}{L}, \tag{13}$$

where $\text{LCS}(\cdot)$ is the length of the longest common subsequence. In our case, it represents the longest sequence of tokens that appear in the same order in both contexts, despite irrelevant context or order changes. The score $\text{DAS}(\hat{G}_{\mathbf{z}_i^0}^{\text{Left}}, \hat{G}_{\mathbf{z}_i^0}^{s_1})$ measures how the semantic dependency changes when appending irrelevant context $\mathbf{z}^{0(s_2)}$ to the left of the original sequence $\mathbf{z}^{0(s_1)}$. Similar $\text{DAS}(\hat{G}_{\mathbf{z}_i^0}^{\text{Right}}, \hat{G}_{\mathbf{z}_i^0}^{s_1})$ can be obtained, which measures the changes of semantic dependency when appending irrelevant context $\mathbf{z}^{0(s_2)}$ to the right.

**Semantic Dependency Analysis with Irrelevant Context Order Change**   For irrelevant context order change, we observe whether the token dependency in sentence *"The sky is blue."* alters when inputting the sentence with irrelevant context order change, e.g., *"The sky is blue. The apple is red."* and input *"The apple is red. The sky is blue."*. We simply use $\text{DAS}(\hat{G}_{\mathbf{z}_i^0}^{\text{Left}}, \hat{G}_{\mathbf{z}_i^0}^{\text{Right}})$ to measure how the semantic dependency changes when appending the irrelevant context $\mathbf{z}^{0(s_2)}$ to the left and the right of the original sequence $\mathbf{z}^{0(s_1)}$.

**Experiments**   We conducted the semantic dependency analysis across over 5,000 cases to examine the impact of irrelevant context added to both the left and right sides, as well as the effect of sequence order changes, in order to determine whether the semantic dependency encoding is context-dependent and order-dependent. Specifically, we measured the dependency changes when perturbing the token $\mathbf{z}_i^{0(s_1)}$ in the original sequence $\mathbf{z}^{0(s_1)}$. This involves evaluating the dependency alterations of its semantically dependent token groups by aligning the top 5 semantically dependent token groups ($L = 5$) and by aligning all tokens from the original sequence $\mathbf{z}^{0(s_1)}$ ($L = N_1$). The average dependency alteration scores are presented in Figure 7.

**Results**   Figure 7(a) and Figure 7(b) illustrate the changes in semantic dependency when irrelevant context is appended on the left or right side. It shows that the rank of semantic dependency strength of common token is significantly affected by the context, while relationships of semantically more related tokens (Top 5) remain relatively stable.

Figure 7(c) further compares the changes in dependency when the irrelevant context is added to the left versus the right side of the original sentence. The results reveal that adding context to the left side generally results in a greater alteration of

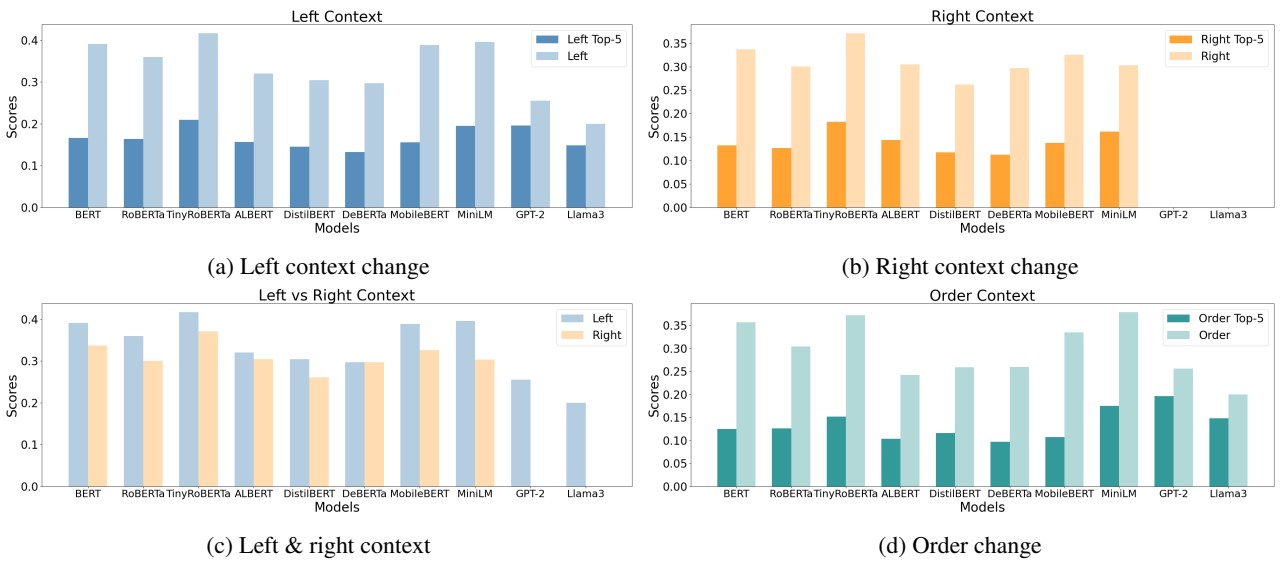

(a) Left context change

(b) Right context change

(c) Left & right context

(d) Order change

Figure 7: Semantic Dependency Alteration Score when irrelevant context or context order changes.

semantic dependencies compared to the right side. This suggests that the order of irrelevant context can differentially impact the model's semantic dependency structures.

Figure 7(d) demonstrates the impact of altering the sequence order on semantic dependencies. The results also show that irrelevant token groups are easily influenced by unrelated contexts, while semantically more dependent tokens exhibit greater resilience to such alterations.

Overall, our findings indicate that both the introduction of irrelevant context and the modification of sequence order dramatically influence semantic dependency within sentences. These results reinforce the importance of context placement and order in shaping the semantic dependency structures learned by Transformer-based language models.

**Insight for Future Model Design**    Our insight can further help training or finetuning a robust language model. Intuitively, semantic dependencies between tokens should remain robust regardless of changes in irrelevant contexts or the order of independent sentences. A natural thought for future work could be regulating transformer models to maintain consistent semantic dependencies despite irrelevant context variations. This may involve implementing regularization techniques that enforce stable token representations regardless of irrelevant context or sequence alterations.

### A.3. More Experiments

**Percentage of a Token Primarily Retains Its Original Semantic Information across Different Datasets in Section 3.**    We measure the total percentage with various sentences from six datasets, including gsm8k (Cobbe et al., 2021), Yelp (Zhang et al., 2015), GLUE (Wang et al., 2019), CNN/DailyMail (Hermann et al., 2015), OpenOrca (Lian et al., 2023) and WikiText (Merity et al., 2016). For each model, over 600,000 token cases were evaluated (each token perturbation is treated as one case). We also observed that a proportion of tokens in GPT propagate semantic information mostly to its next token. Thus, We also include the percentage of the token propagating semantic information to both its next token and itself for GPT on the right. The detailed result is displayed in Table 5.

To further discuss this phenomenon, we believe that one key factor is the presence of residual shortcuts, which may encourage final-layer token representations to retain information from the original input. For example, in a simple one-layer model with a direct shortcut from input to output, the final-layer token is likely to closely mirror its corresponding input token. However, residual connections alone do not fully explain the observed effect. To support this, we include the percentages for GPT-2, GPT-2-Large, and GPT-2-XL across different datasets in Table 6. Although these models share the same residual architecture, larger models (e.g., GPT-2-XL) exhibit significantly stronger semantic retention (similar percentages to BERT, e.g., around 98%) at the final-layer token level than smaller models (e.g., GPT-2). It suggests that semantic retention is also influenced by other factors such as model size and complexity.

Table 5: Percentage of a token primarily retains its original semantic information.

|  | gsm8k | Yelp | GLUE | DailyMail | OpenOrca | WikiText |
|---|---|---|---|---|---|---|
| BERT | 99.22 | 98.58 | 98.48 | 98.81 | 98.90 | 98.84 |
| RoBERTa | 89.54 | 93.46 | 89.98 | 95.02 | 93.34 | 96.99 |
| TinyRoBERTa | 92.29 | 95.16 | 94.43 | 95.11 | 94.38 | 94.39 |
| ALBERT | 96.84 | 97.36 | 97.67 | 96.67 | 97.65 | 95.85 |
| DistilBERT | 93.84 | 95.27 | 95.84 | 95.70 | 95.54 | 94.49 |
| DeBERTa | 99.69 | 99.57 | 99.54 | 99.67 | 99.46 | 99.78 |
| MobileBERT | 94.34 | 96.38 | 93.16 | 97.73 | 98.22 | 99.08 |
| MiniLM | 87.21 | 92.16 | 93.25 | 87.54 | 86.38 | 85.58 |
| GPT-2 | 75.19/88.42 | 77.46/89.94 | 77.49/92.51 | 73.11/85.88 | 69.32/81.68 | 72.31/84.46 |
| LLaMA3 | 96.21 | 96.68 | 94.20 | 95.85 | 95.78 | 94.80 |

Table 6: Percentage of a token primarily retains its original semantic information in GPT series.

|  | gsm8k | Yelp | GLUE | DailyMail | OpenOrca | WikiText | Avg. Percentage |
|---|---|---|---|---|---|---|---|
| GPT-2 (124M) | 75.19 | 77.46 | 77.49 | 73.11 | 69.32 | 72.31 | 75.15 |
| GPT-2-Large (774M) | 98.49 | 98.47 | 98.16 | 98.17 | 98.34 | 98.08 | 98.29 |
| GPT-2-XL (1.5B) | 98.64 | 98.32 | 97.85 | 97.83 | 97.90 | 97.80 | 98.05 |

**Percentage of a Token Propagates Semantic Information to Other Tokens in Section 3.** We also observe the change of the specific input word causes influence on other tokens in the final layer in experiment of Section 3. The result in all cases (each token perturbation is treated as one case, over 600,000 cases are evaluated for each model) is shown in Table 7. Even if minor, in models like BERT, DeBERTa, and MobileBERT, the change is almost 100%, which means each token receives pieces of semantic information from almost every token in the input sequence. While in auto-regressive models like LLaMA or GPT, the token only influences the tokens on this token's right side. We observe the changes of tokens on each tokens' left side is 0. We can also observe the change exists in all tokens on each token's right side, which suggests each token receives pieces of semantic information from almost all tokens on its left side.

**Why replacing a token with random tokens to explore semantic dependency** For both finding 2 and finding 3, we need to examine how semantic dependency is encoded in the final layer by replacing an input token $z_i^0$ and observing which final-layer token (e.g., at position $j$) changes. If $z_i^0$ and $z_j^L$ exhibit strong dependency, which means the semantic information of $z_i^0$ is encoded in the final-layer representation at position $j$, then replacing $z_i^0$ with another token $\tilde{z}_i^0$ should cause $z_j^L$ to change substantially. This indicates a semantic dependency between the two tokens. However, if we replace $z_i^0$ with a synonym (e.g., $z_i^{'0}$), the overall semantic meaning of the sentence may remain largely unchanged, and the model may treat $z_i^0$ and $z_i^{'0}$ similarly. In this case, we may observe a minimal change at $z_j^L$, making it difficult to conclude whether $z_j^L$ was originally dependent on $z_i^0$, even if a true dependency existed. Therefore, we use random tokens to encourage semantic independence. It is also important to note that random token selection may introduce out-of-domain predictions. We believe measuring sensitivity or semantic relevance through gradients could provide valuable insights and exciting directions for future work.

**Why Using Neural Dependency Parsing Tool in Section 4** Noted that our analysis relies on semantic dependency data derived with SpaCy, a pretrained neural network-based dependency parser. SpaCy generates syntactic dependency trees using robust neural architectures trained on large annotated corpora, offering a reliable approximation of semantic dependencies. To our knowledge, no token-level semantic dependency dataset with comprehensive human annotations exists. Constructing such a dataset would be prohibitively expensive and prone to omissions due to the complexity of identifying all dependent token relationships manually. Thus, we use neural dependency parsing tool to generate a specialized semantic dependency dataset for our experiment.

We additionally conducted experiments using another widely adopted dependency parser, Stanza (Stanford NLP) (Qi et al., 2020), to validate the robustness of our findings. As shown in Table 8, the results obtained with Stanza are consistent with those derived from SpaCy, further supporting the conclusion that transformer models encode truthful semantic dependencies

Table 7: Percentage of a token propagates semantic information to other tokens.

| | gsm8k | Yelp | GLUE | DailyMail | OpenOrca | WikiText |
|---|---|---|---|---|---|---|
| BERT | 99.44 | 99.09 | 99.16 | 99.20 | 99.34 | 99.52 |
| RoBERTa | 92.94 | 95.00 | 90.76 | 95.98 | 95.23 | 98.24 |
| TinyRoBERTa | 96.46 | 96.42 | 97.04 | 96.42 | 95.98 | 96.07 |
| ALBERT | 97.88 | 97.99 | 98.35 | 97.34 | 98.23 | 96.63 |
| DistilBERT | 95.44 | 95.89 | 96.37 | 96.29 | 96.42 | 95.93 |
| DeBERTa | 99.86 | 99.63 | 99.66 | 99.77 | 99.68 | 99.82 |
| MobileBERT | 99.43 | 99.22 | 98.96 | 99.38 | 99.50 | 99.74 |
| MiniLM | 94.03 | 95.51 | 94.71 | 91.74 | 92.19 | 92.35 |
| GPT-2 | 100.00 | 100.00 | 100.00 | 100.00 | 100.00 | 100.00 |
| LLaMA3 | 100.00 | 100.00 | 100.00 | 100.00 | 100.00 | 100.00 |

Table 8: Alignment scores indicating how well tokens encode truthful semantic dependencies using Stanza and Spacy (%).

| | BERT | RoBERTa | tinyRoBERTa | ALBERT | DistilBERT | DeBERTa | MobileBERT | MiniLM | GPT-2 | LLaMA3 |
|---|---|---|---|---|---|---|---|---|---|---|
| SpaCy | 87.86 | 87.71 | 82.44 | 88.77 | 88.88 | 87.17 | 85.8 | 84.62 | 93.41 | 92.47 |
| Stanza | 84.33 | 86.9 | 81.14 | 85.53 | 87.19 | 83.69 | 80.98 | 83.67 | 91.42 | 90.32 |

in their final layers.

Note that although model-estimated semantic dependencies can be easily obtained, the main challenge is that existing semantic dependency parser methods usually cannot measure dependencies at the subword level. This makes direct comparison difficult. To address this issue, we may need to manually annotate the semantic dependencies and compare them with those estimated by the models, which is costly and hard to scale.

**Why Using Longest Common Subsequence in Section A.2** Consider a simple example to understand how LCS captures changes in token order: Suppose we have two sequences, $A = [1, 2, 3, 4]$ and $B = [2, 3, 4, 1]$. In moving from sequence $A$ to sequence $B$, the order of the tokens changes such that the token "1" moves from the beginning to the end. Here, the LCS between $A$ and $B$ is the subsequence $[2, 3, 4]$, which has a length of 3. This subsequence represents the largest set of tokens that have retained their original order between the two sequences. Since the total number of tokens, $N$, is 4, the LCS length of 3 indicates that one token ("1") changed its position relative to the others. By calculating DAS $= 0.25$, we find that a quarter of the token order has been altered due to the change in context. Thus, a lower LCS value (relative to $N$) results in a higher DAS, reflecting a more significant change in token dependency patterns. This metric effectively highlights how sensitive the token dependencies are to contextual modifications, demonstrating the dynamic nature of semantic processing in natural language systems.

**Why Choosing QA as the Primary Task for Our Experiments** We chose the question-answering (QA) task because it is particularly well-suited for evaluating the impact of semantic dependency mistakes at the token level. QA tasks inherently involve understanding and associating tokens in a question with those in the context, making them ideal for testing the model's ability to handle complex dependencies. This directly aligns with the focus of our study, which explores how false encoded semantic dependencies lead to model mistakes. Additionally, to validate our findings, it is crucial to have ground truth datasets that clearly present correct and incorrect dependencies. QA tasks provide datasets like SQuAD, where the answers are explicitly tied to certain context tokens. These datasets enable us to systematically evaluate how dependency mistakes between question and context tokens contribute to prediction mistakes.

**Why Threshold of F1 $< 0.6$ is Chosen and Additional Evaluation Using More Advanced ChatGPT Model.** The threshold of F1 $< 0.6$ for identifying incorrect answers was determined empirically. Since our goal is to assess whether incorrect answers are associated with incorrect semantic dependencies, an F1 score below 0.6 indicates that over 40% of the tokens predicted by the model differ from those in the original answer, which strongly suggests the answer is likely incorrect.

To further strengthen this analysis, and following existing work, we conducted additional experiments using ChatGPT-4o

Table 9: Additional experiments ChatGPT-4o model to find incorrect cases.

|  | BERT | RoBERTa | tinyRoBERTa | ALBERT | DistilBERT | DeBERTa | MobileBERT | MiniLM | GPT-2 | LLaMA3 |
|---|---|---|---|---|---|---|---|---|---|---|
| $P(f_\theta)$ (F1<0.6) | 79.07 | 69.20 | 77.94 | 71.86 | 81.80 | 75.32 | 66.61 | 77.56 | 48.04 | 64.56 |
| F1 Score | 92.93 | 84.86 | 82.83 | 80.56 | 85.71 | 91.69 | 81.19 | 85.34 | 0.78 | 35.81 |
| $P(f_\theta)$ (GPT-4o select) | 79.00 | 68.42 | 73.31 | 66.11 | 81.79 | 77.84 | 68.69 | 77.56 | 59.60 | 62.35 |
| Accuracy | 88.45 | 78.00 | 78.00 | 74.63 | 76.63 | 90.44 | 74.50 | 78.90 | 0.10 | 14.68 |

Table 10: The F1 score of GPT and LLama models using one-shot setting and zero-shot setting.

|  | F1 (0-shot) | F1 (1-shot) |
|---|---|---|
| GPT-2 (124M) | 0.78 | 5.5 |
| GPT-2-Large (774M) | 7.3 | 21.09 |
| LLaMA3-8B-instruct (8B) | 35.81 | 76.27 |

models to compare the model's answer with the ground truth and find incorrect cases. The results displayed in Table 9 are similar to using F1<0.6.

**Why Choose 0-shot F1 Evaluation on LLaMA and GPT Models** To ensure a fair comparison, we evaluated LLaMA and GPT models using the same zero-shot (0-shot) setting as BERT. This is the reason that they present a low accuracy. Table 10 shows the F1 score using a one-shot setting, which aligns with official benchmark evaluations.

## A.4. Localize Parameters of Attention Head Group Responsible for Semantic Dependency in Section 5.2

To better understand the network's role in model mistakes and evaluate whether false semantic dependencies can be adjusted to improve model performance, we focus on localizing the parameters that encode semantic dependencies. We have developed a method to identify the attention heads primarily responsible for specific token dependencies. Here, we present the intuition and detailed equations.

Intuitively, when the input token carrying specific semantic information changes, the attention heads relevant to corresponding semantic information propagation will exhibit significant changes in their outputs, while the outputs of irrelevant heads will remain relatively unchanged. Therefore, by identifying heads with the highest variation in their contribution to a given token dependency, we can pinpoint the group of attention heads that are mutually responsible for any token dependency including wrong or correct token dependency in the QA task.

As mentioned in Eq. (2), transformer encoder or transformer decoder is a residual network built from $L$ layers, each of which contains a multi-head self-attention (MHA) followed by feed forward network (FFN) block.

In the $l$-th MHA layer, the input stream $z^{l-1}$ is processed separately by $H$ attention heads. Specifically, the input sequence $Z^{l-1}$ is separately projected into $Q$, $K$, $V$ matrix in $h$-th attention head of $l$-th layer as follows:

$$\mathbf{Q}^{l,h} = \mathbf{Z}^{l-1}\mathbf{W}_Q^{l,h}, \quad \mathbf{K}^{l,h} = \mathbf{Z}^{l-1}\mathbf{W}_K^{l,h}, \quad \mathbf{V}^{l,h} = \mathbf{Z}^{l-1}\mathbf{W}_V^{l,h} \tag{14}$$

Then attention weight matrix $\mathbf{A}^{l,h} \in \mathbb{R}^{N \times N}$ is calculated as follows:

$$\mathbf{A}^{l,h} = \text{softmax}\left(\frac{\mathbf{Q}\mathbf{K}^T}{\sqrt{d_k}}\right) \tag{15}$$

The output of each attention head is

$$\mathbf{O}^{l,h} = \mathbf{A}^{l,h}\mathbf{V}^{l,h} \tag{16}$$

For multi-head attention, the outputs of each head are concated and projected to $Z^l \in \mathbb{R}^{N \times D}$, where $W_O$ is the output weight matrix.

$$\mathbf{MHA}^l(\mathbf{z}^{l-1}) = \text{Concat}(\mathbf{O}^{l,1}, \mathbf{O}^{l,2}, \ldots, \mathbf{O}^{l,H})\mathbf{W}_O \tag{17}$$

The class token and the other tokens share the same computation process. Inspired by previous study (Gandelsman et al., 2024), the contribution of $l$-th MHA on $j$-th token can be broken down into tokens and heads. We can observe that given a token, each context token contributes to this token by adding operation for semantic information aggregation, which generates context-related token representation.

$$\text{MHA}_j^l(\mathbf{Z}^{l-1}) = \sum_{h=1}^{H} \sum_{i=1}^{N} x_i^{l,h}, \quad x_i^{l,h} = \alpha_i^{l,h} W_{VO}^{l,h} z_i^{l-1} \tag{18}$$

Specifically, for any token dependency, i.e., token dependency from $i$-th token to $j$-th token, including correct or wrong token dependency in the QA task mentioned above, we replace the $i$-th token with $K$ randomly sampled tokens. Then we measure each head's contribution to semantic dependency by calculating average change $\Delta_{\mathbf{z}_j^L | \mathbf{z}_i^0}^{l,h}$ between original head contribution and perturbed head contributions as follows:

$$\Delta_{\mathbf{z}_j^L | \mathbf{z}_i^0}^{l,h} = \frac{1}{K} \sum_{k=1}^{K} \left\| x_i^{l,h(k)} - x_i^{l,h(\text{org})} \right\|_2 \tag{19}$$

As is shown in Figure 4(b), we test the dependency contribution score $\Delta_{\mathbf{q}_j^L | \mathbf{a}_i^0}^{l,h}$ of each attention head in BERT for both wrong semantic dependency between "sign" and "?" and correct semantic dependency between "anthem" to "marry" in corresponding QA instance. In this case we can observe there are a group of attention heads (highlighted with bright color in the contribution heatmap) mutually contribute to the semantic dependency. We can also find the head group responsible for false dependency is clearly brighter than correct dependency, showing a different pattern.

**Extra Experiments and Results**   Firstly, we calculated the average number of top 5% contributing attention heads per layer for both correct and false semantic dependency across all BERT's failed QA cases (shown in Figure 8). The results reveal the heads responsible for false semantic dependency are primarily distributed in later layers, whereas those contributing to correct semantic dependency are distributed across both earlier and later layers.

Secondly, We have analyzed the frequency of each top 5% contributing attention head for correct and false semantic dependency across all failed QA cases (a part of the result is shown in the main text in Figure 5). Here, we include all fine-tuned BERT models commonly used in QA tasks with high accuracy (F1 > 0.8) in Figure 9.

Additionally, We also calculated the average number of top 5% contributing attention heads per Layer for semantic dependency across successful QA cases in BERT series (the same number of failed QA cases randomly sampled from the SQuAD dataset) in Figure 10. The results show that when the model performs correctly, the heads responsible for the correct semantic dependency are mostly distributed in later layers compared to correct semantic dependency when the model fails a QA task. Based on Figure 8 and Figure 10, we surmise that The model tends to rely more on the attention heads in the later layers when answering questions, regardless of whether the answer is correct or incorrect.

Furthermore, we also count the frequency of each top 5% contributing attention head across successful QA cases (the same number of failed QA cases randomly sampled from the SQuAD dataset) in Figure 11. We also found a similar group of top attention heads responsible for correct semantic dependency when models fail or succeed in QA tasks, which means they are QA task-specific heads. The high overlap of the task-specific attention heads also shows that the same parameters contribute to encoding correct semantic dependency regardless of whether models succeed or fail in QA cases. Based on Figure 9 and Figure 11, we can conclude that such model mistakes in QA tasks can not be corrected by directly adjusting parameters such as simply pruning attention heads because correct and incorrect dependencies are controlled by the same group of parameters. We need other strategies to address the challenge of removing false dependencies without affecting the correct ones.

**Discussion**   As highlighted in the main text, the attention mechanism plays a key role in propagating semantic information between tokens, ultimately enabling final layer tokens to encode semantic dependencies. In this paper, we focus primarily on analyzing the contribution of attention head parameters. Additionally, we notice that MLP layers may amplify irrelevant or erroneous semantics (Geva et al., 2023). In future work, we aim to extend our analysis to quantify the contribution of MLP layers to semantic dependency.

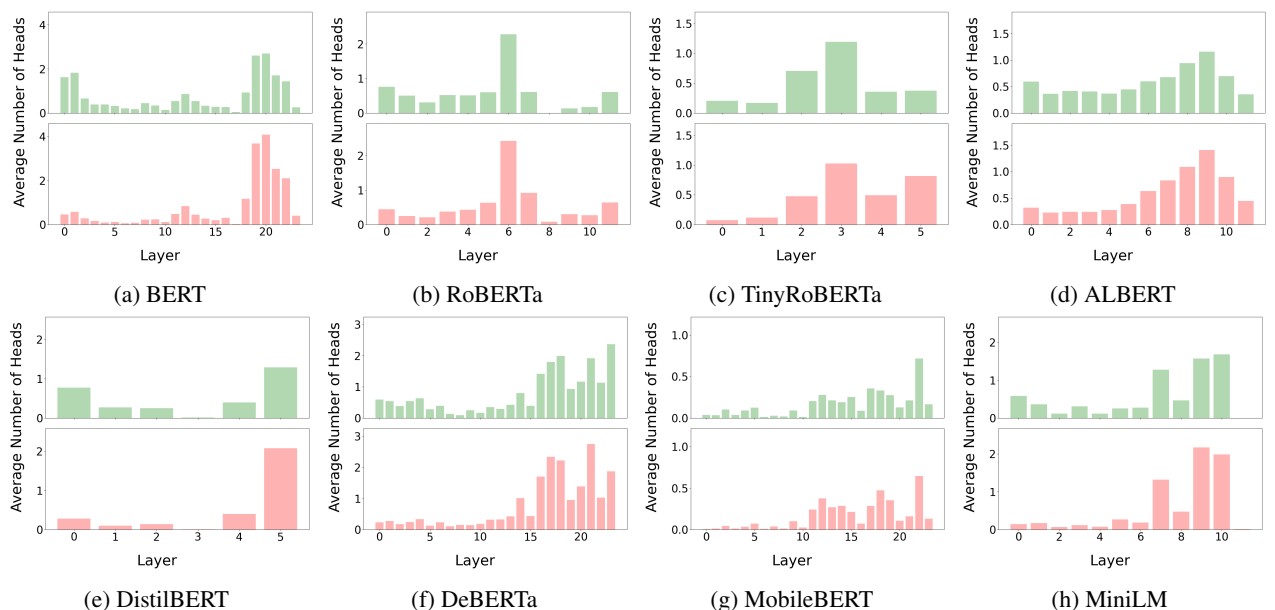

Figure 8: Distribution of top 5% contributing attention heads for correct (green) and false (red) semantic dependency across all failed QA cases.

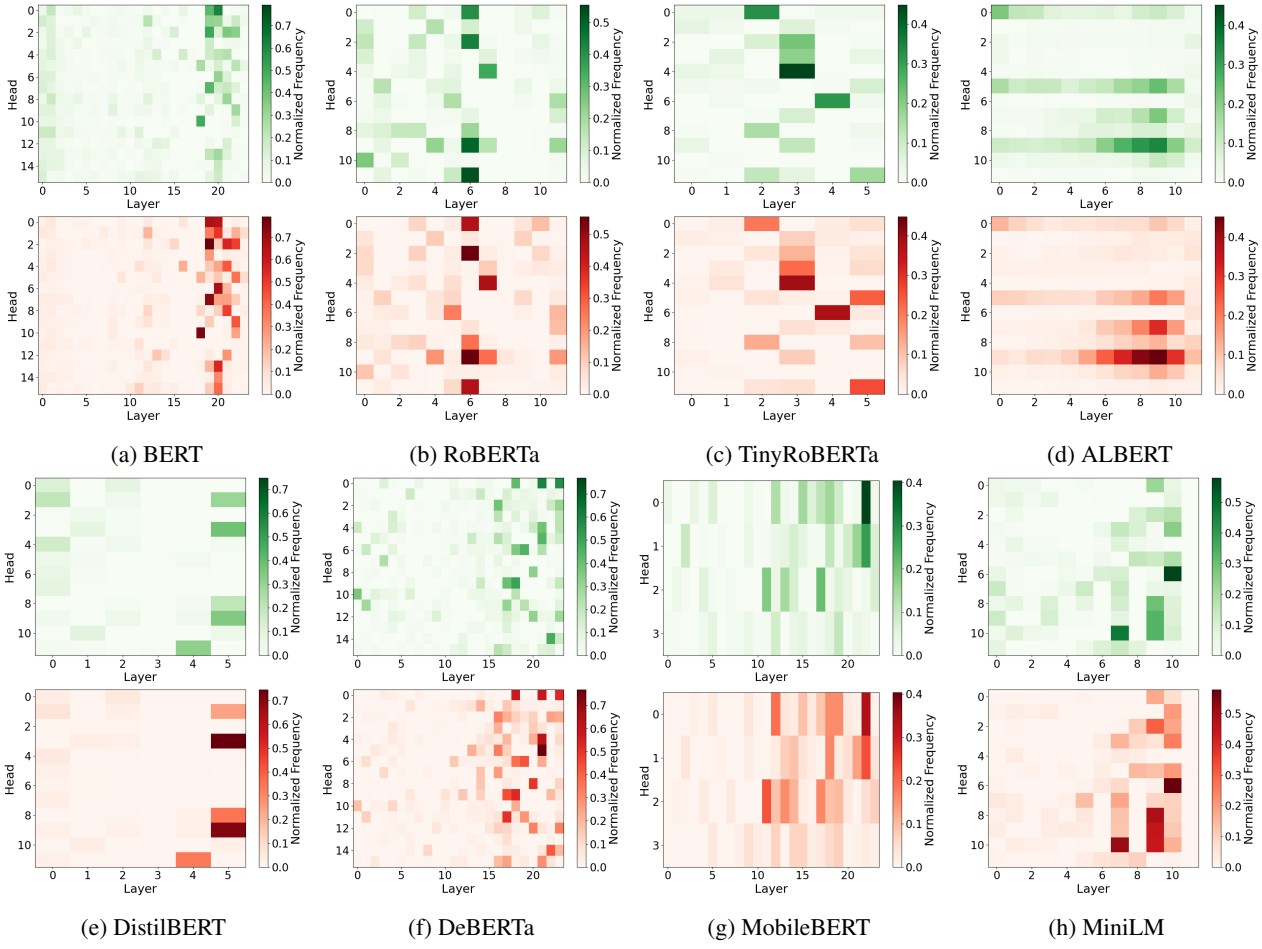

Figure 9: Frequency of each top 5% contributing attention head for correct (green) and false (red) semantic dependency across all failed QA cases.

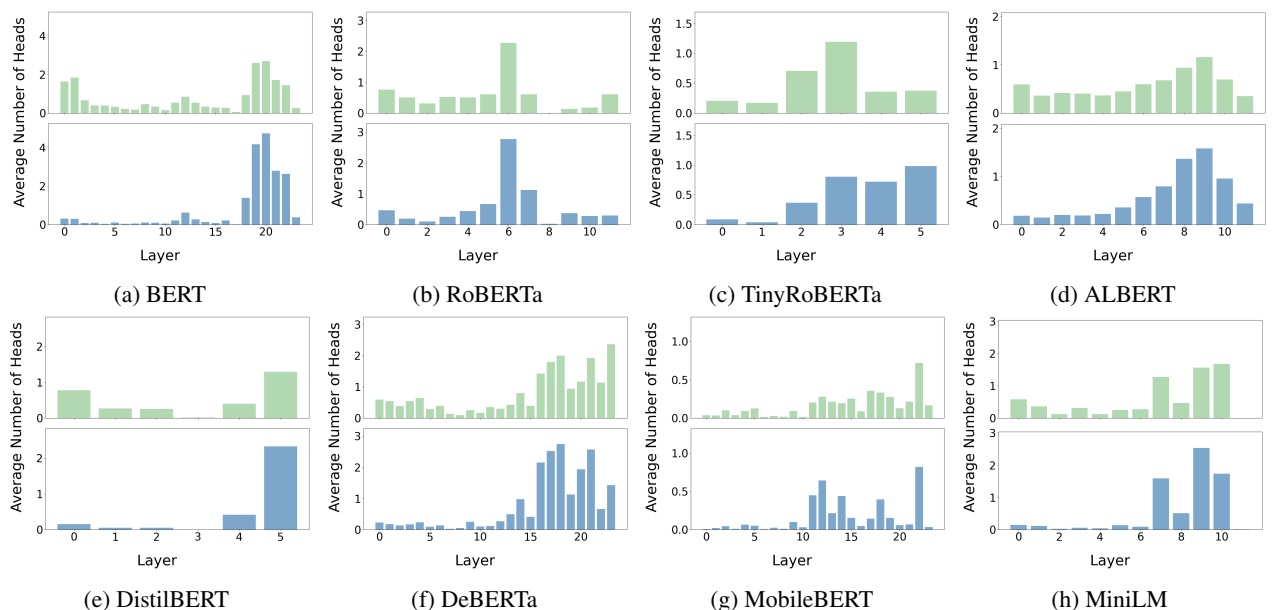

Figure 10: Distribution of top 5% contributing attention heads for correct semantic dependency in failed QA cases (green) compared to successful QA cases (blue).

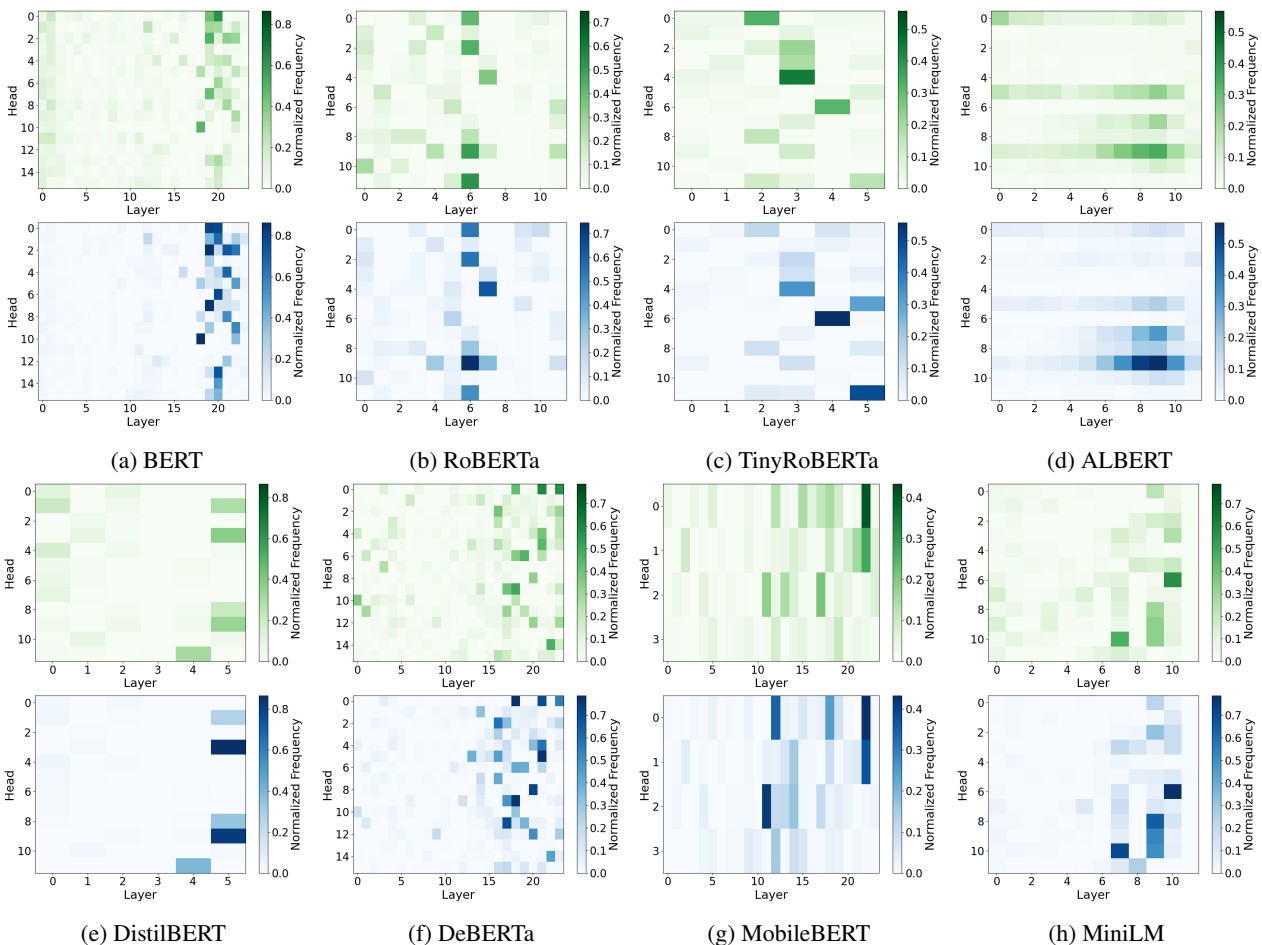

Figure 11: Frequency of each top 5% contributing attention head for correct semantic dependency in failed QA cases (green) compared to successful QA cases (blue).

## A.5. Pesudocode for Section 5

---

**Algorithm 1** Evaluation of Semantic Dependencies

---

**Input:** Dataset with $M$ instances, transformer model $f_\theta$, number of perturbations $K$.

**Output:** The percentage $p$ that $\Delta'_{A_{\text{wrong}}|Q}$ is greater than $\Delta'_{A_{\text{correct}}|Q}$ given the question and answer pairs where the model makes mistakes.

Initialize $count \leftarrow 0$

**for** each incorrect QA instance $m = 1$ **to** $H$ **do**

   Extract question tokens $Q = \{\mathbf{q}_i^0\}_{i=1}^{N_Q}$, correct answer tokens $A_{\text{correct}} = \{\mathbf{a}_i^0\}_{i=1}^{N_C}$, and incorrect answer tokens $A_{\text{wrong}} = \{\mathbf{a}_i^0\}_{i=1}^{N_W}$

   **for** each answer token $\mathbf{a}_k^0 \in A_{\text{correct}} \cup A_{\text{wrong}}$ **do**

      **for** $k = 1$ **to** $K$ **do**

         **if** $k = 1$ **then**

            $\tilde{\mathbf{z}}_k^0 \leftarrow \mathbf{a}_k^0$

         **else**

            $\tilde{\mathbf{z}}_k^0 \leftarrow \text{RandomToken}(\mathcal{V})$

         **end if**

         Construct perturbed sequence $\tilde{\mathbf{z}}^{0(k)}$ by replacing $\mathbf{a}_k^0$ with $\tilde{\mathbf{z}}_k^0$

         Compute final layer representations $\tilde{\mathbf{z}}^{L(k)} \leftarrow f_\theta(\tilde{\mathbf{z}}^{0(k)})$

      **end for**

      Compute original final layer representations $\mathbf{z}^{L(\text{org})} \leftarrow f_\theta(\mathbf{z}^{0(\text{org})})$

      **for** each token $j = 1$ **to** $N$ **do**

         Calculate $\Delta_{\mathbf{z}_j^L|\mathbf{a}_k^0} \leftarrow \frac{1}{K-1}\sum_{k=2}^{K}\left\|\tilde{\mathbf{z}}_j^{L(k)} - \mathbf{z}_j^{L(\text{org})}\right\|_2$

      **end for**

      Determine maximum dependency score for $\mathbf{a}_k^0$: $\Delta'_{\mathbf{a}_k^0|Q} = \max_{j=1}^{N_Q} \Delta_{\mathbf{q}_j^L|\mathbf{a}_k^0}$

   **end for**

   Determine maximum dependency score for correct answers: $\Delta'_{A_{\text{correct}}|Q} = \max_{k=1}^{N_C} \Delta'_{\mathbf{a}_k^0}$

   Determine maximum dependency score for wrong answers: $\Delta'_{A_{\text{wrong}}|Q} = \max_{k=1}^{N_W} \Delta'_{\mathbf{a}_k^0}$

   **if** $\Delta'_{A_{\text{wrong}}} > \Delta'_{A_{\text{correct}}}$ **then**

      $count \leftarrow count + 1$

   **end if**

**end for**

Calculate percentage: $p(f_\theta) = \frac{count}{M}$

**Return:** $p(f_\theta)$

---

## A.6. Symbol List

Table 11: Symbols and Their Explanations

| Symbol | Explanation |
|---|---|
| $\mathbf{z}_i^l$ | The embedding of the $i$-th token in the $l$-th layer. |
| $\mathbf{z}_j^l$ | The embedding of the $j$-th token in the $l$-th layer. |
| $\mathbf{z}_i^{l\,(\mathrm{org})}$ | The original embedding of the $i$-th token in the $l$-th layer. |
| $\tilde{\mathbf{z}}_i^{L(k)}$ | The $k$-th perturbed embedding of the $i$-th token in the $l$-th layer. |
| $\Delta_{\mathbf{z}_j^L \mid \mathbf{z}_i^0}$ | Semantic dependency score, which measures how the perturbation of token $i$ at layer 0 affects token $j$ at the final layer $L$. |
| $N$ | The number of tokens in a token sequence. |
| $K$ | The $i$-th token in layer 0 is perturbed $K$ times to calculate the average change of the $i$-th token in layer $L$. $K = 5$ in our experiments. |
| $M$ | The number of total perturbed token cases across all sequences we evaluate. |
| $P(f_\theta)$ | Percentage $P$ of the cases that the transformer-based language model $f_\theta$ matches our finding. |
| $W_{\mathbf{w}_i^0}$ | True semantically dependent word group for the $i$-th word in layer 0 based on semantic dependency parsing. |
| $G_{\mathbf{z}_i^0}$ | Truthful semantically dependent token group for the $i$-th token in layer 0 based on semantic dependency parsing. |
| $\hat{G}_{\mathbf{z}_i^0}$ | Estimated semantically dependent token group for the $i$-th token using token perturbation. |
| $K_{\mathrm{top}}$ | The number of top tokens most sensitive to the perturbation of the input token. $K_{\mathrm{top}}$ is set to the size of $G_{\mathbf{z}_i^0}$. In the experiment, we evaluate the overlap of $G_{\mathbf{z}_i^0}$ and top 5 tokens when the size is under 5. |
| $S_{z_i^0}$ | Alignment score between the truthful ($G_{z_i^0}$) and estimated ($\hat{G}_{z_i^0}$) semantically dependent token groups. |
| $\hat{G}_{\mathbf{z}_i^0}^{s_1}, \hat{G}_{\mathbf{z}_i^0}^{s_2}$ | Estimated semantically dependent token group for the $i$-th token corresponding to token sequences $s_1$ and $s_2$. |
| $\hat{G}_{\mathbf{z}_i^0}^{\mathrm{Left}}, \hat{G}_{\mathbf{z}_i^0}^{\mathrm{Right}}$ | Estimated semantically dependent token group for the $i$-th token corresponding to concatenated sequences $(s_2, s_1)$ and $(s_1, s_2)$. |
| $\mathrm{DAS}(\cdot)$ | Dependency Alteration Score, measuring the impact of irrelevant context or sequence order changes on semantic dependencies in a sequence. |
| $L$ | The number of chosen semantically dependent tokens in the original token sequence $\mathbf{z}^{0(s_1)}$. e.g., $L = 5$ when choosing the top 5 semantically dependent tokens for evaluation. |
| $\mathbf{q}_i^l$ | The embedding of the $i$-th question token in the $l$-th layer. |
| $\mathbf{a}_i^l$ | The embedding of the $j$-th answer token in the $l$-th layer. |
| $\Delta_{\mathbf{q}_j^L \mid \mathbf{a}_i^0}$ | Semantic dependency score in QA task, which measures how the perturbation of $i$-th answer token at layer 0 affects $j$-th question token at the final layer $L$. |
| $\Delta'_{a_i^0 \mid Q}$ | Highest semantic dependency score above all semantic dependency between all question tokens and $i$-th answer tokens in a QA task. |
| $\Delta'_{A_{\mathrm{correct}} \mid Q}, \Delta'_{A_{\mathrm{wrong}} \mid Q}$ | Highest semantic dependency score above all semantic dependency between question tokens and answer tokens (correct or wrong) in a QA task. |

