# OpenReview forum: "A Lens into Interpretable Transformer Mistakes via Semantic Dependency"
_ICML.cc/2025/Conference — ICML 2025 poster_

### Official Review · Reviewer_1yCg · 2025-03-08

**Overall Recommendation:** 2

**Summary:**

The paper proposes a score for measuring "semantic dependency" of final token activations on various input tokens. Specifically, for a final layer token activation at token $j$, and a token $i$ in the same input sequence, the semantic dependency is the expected euclidean norm of the the change in final layer token activation at $j$ when $i$ is counterfactually perturbed to a random token.

The authors then measure various properties of their score, namely:
- Showing that final layer activation is most dependent on its own token
- Correlating their semantic dependency score with semantic dependency groupings from SpaCy
- Correlating their semantic dependency score with failures to answer QA tasks

Lastly, the authors attributed how each attention head outputting at token $j$ contributes to semantic dependency when token $i$ is changed.

## Update after rebuttal
I do not think the rebuttal addressed the core issues that I raised in my initial review. For example, the authors did not make a convincing case that their experiments are in any way causal; to show causality one needs to causally intervene on semantic dependency. In addition, the authors did not provide any additional empirical evidence to address the limitations of the original experiments. I have therefore chosen to not increase my recommendation.

**Claims And Evidence:**

The authors use causal language like "mistakes arise from the model’s tendency to encode false semantic dependency in tokens through transformer layers", and "model mistakes in QA tasks stem from incorrect semantic dependencies en-coded in question tokens". However, the experiments in section 5 are correlational.

**Essential References Not Discussed:**

The idea of making token level ablations to causally identify dependencies has a long history that is not engaged with at all in the paper (Vig et al, 2020; Finlayson et al, 2021; Amini et al 2022).

The form of semantic dependency discussed in the paper is also related to the study of relational binding (Wattenberg et al, 2024; Feng et al, 2023)

Finlayson, Matthew, et al. "Causal analysis of syntactic agreement mechanisms in neural language models." arXiv preprint arXiv:2106.06087 (2021).

Vig, Jesse, et al. "Investigating gender bias in language models using causal mediation analysis." Advances in neural information processing systems 33 (2020): 12388-12401.

Amini, Afra, et al. "Naturalistic causal probing for morpho-syntax." Transactions of the Association for Computational Linguistics 11 (2023): 384-403.

Wattenberg, Martin, and Fernanda B. Viégas. "Relational composition in neural networks: A survey and call to action." arXiv preprint arXiv:2407.14662 (2024).

Feng, Jiahai, and Jacob Steinhardt. "How do language models bind entities in context?." arXiv preprint arXiv:2310.17191 (2023).

**Experimental Designs Or Analyses:**

- Validation 2 in Table 1 is extremely sensitive. The quantity $\Delta$ is an average of norms, which is guaranteed to be non-negative. The quantity measures the proportion of pairs of token for which the $\Delta$ is positive, which doesn't seem very informative; by default I'd expect the vast majority of tokens to exhibit _some_ dependence.

- The evaluation in Table 3 also is missing base rates. Specifically, the authors only showed that a large fraction of incorrect answers also have dependency scores higher on the wrong answer tokens compared to the correct answer tokens. However, this needs to be compared to the base rate of the latter happening. I suggest using something like spearman's rho, or even better, show the rates of all two by two possibilities.

- Why is the F1 score of llama-3 in table 3 so poor compared to all the other models? It is by far the best model in the table according to standard benchmarks such as MMLU.

**Methods And Evaluation Criteria:**

N/A

**Other Comments Or Suggestions:**

- typo in line 199, second column. repeated subscript $i$, one of them should be $j$.
- same place: presumably, you meant $i \le j$ for autoregressive models, and not for all $i, j$.

**Other Strengths And Weaknesses:**

N/A

**Questions For Authors:**

N/A

**Relation To Broader Scientific Literature:**

Understanding how LMs represent information is an important area of study. However, looking only at the last layer representation ignores a lot of rich phenomena that occurs within the language model. Further, it is not clear to what extent the semantic dependency score says anything at all about the internal representations of the model; consider for instance an alternate score that is defined identically to the proposed score, but measures distances (say using KL) in the log probs rather than the final layer activations. These scores would be exceedingly similar, differing only by the choice of metric (since the logits are just a linear transformation of the scores), but the alternate metric is defined entirely behaviorally, rather than based on any internal representations.

**Theoretical Claims:**

N/A

---

> ### Author Rebuttal · Authors · 2025-04-01
>
> **Q1. (Claims) The authors use causal languageHowever, the experiments in section 5 are correlational.**
>
> **A1.**  Thank you for this insightful question. We believe this question stems from a misunderstanding of what we mean by "cause" in this context. We have revised the paper carefully to make this clearer. Please also kindly let us know of any part that may confuse.  Please kindly allow us to outline the logic of our reasoning:
>
> - **Causal claim:** Encoding incorrect semantic dependencies can lead to incorrect answers. This is a reasonable assumption even for humans. To answer questions correctly, we need to represent semantic dependencies accurately.
> - From the claim, we believe that, for neural networks, semantic dependencies must be somehow encoded in their internal token representations.  Based on this understanding,
> - In Section 4, we observe that in correct predictions, the final-layer token tends to encode semantically dependent words together.
> - In Section 5,  we observe that when semantically dependent words are not encoded together in the final-layer tokens, the model is more likely to make a mistake.
>
>
> ---
>
> **Q2. (Experimental Designs) For Validation 2, I'd expect the vast majority of tokens to exhibit dependence.**
>
> **A2.** Thanks for the insightful questions. This is not a major finding, and we only briefly mention it in a few words in the main text, with the full results presented in the appendix.  Please kindly refer to A3 of our response to Reviewer LWrx for the motivation for highlighting this point.
>
> ---
>
> **Q3. (Experimental Designs) I suggest using all two by two possibilities for Table 3.**
>
> **A3.** We have followed your insightful suggestion and included a table for each model. Due to space limitations, the tables will be attached in the comments.
>
> ---
>
> **Q4. (Experimental Designs)** Why is the F1 score of llama-3 in table 3 so poor compared to all the other models? It is by far the best model in the table according to standard benchmarks such as MMLU.
>
> **A4.** To ensure a fair comparison, we evaluated LLaMA and GPT models using the same zero-shot (0-shot) setting as BERT.  We conducted an additional experiment using a one-shot (1-shot) setting, following the official evaluation method. The performance aligns with official evaluations.
>
> ---
>
> **Q5.  (Broader Literature)  Looking only at the last layer representation ignores a lot of rich phenomena that occur within the language model.**
>
> **A5.** Thank you for the insightful question.
>
> Please note that it is feasible to apply our methods to every layer of tokens. Our motivation for focusing on the final-layer tokens is that we aim to understand errors in the model’s output, and the final layer token should have the greatest influence on the output, as has also been acknowledged in existing works.
>
> We also believe that looking at tokens from other layers can be important, and we would like to explore this further in future work.
>
> ---
>
> **Q6.  (Broader Literature)  It is not clear to what extent the semantic dependency score says anything at all about the internal representations of the model**
>
> **A6.** The purpose of designing the semantic dependency score is to assess whether the model is more likely to make a mistake when semantically dependent words are not jointly encoded in the final-layer token. We believe this finding offers valuable insight into how internal representations relate to model errors, and it can help guide future method development.
>
> ---
>
> **Q7.  (Broader Literature)  Consider an alternate score that uses KL in the log probs rather than the final layer activations.**
>
> **A7.** Thanks for the constructive feedback.
> We believe the score you mentioned may be similar to the method proposed by Feng and Steinhardt (2024).
> However, their method fundamentally differs from ours in its assumptions and goals.
>
> Intuitively, they assume that the model’s most confident output reflects its encoded semantic dependency, and they use KL-based scores to study other downstream properties based on this assumption.
>
> Notably, their KL-based score cannot be used to test the validity of their assumption.
> In contrast, our goal aligns more closely with testing that assumption, specifically, whether there is a statistical dependence between the model’s output and the semantic dependency encoded in the final-layer token. Note that these two lines of research can be complementary.
>
> ---
>
>
> **Q8. (References) token-level ablations to causally identify dependencies are not engaged in the paper.  The semantic dependence is related to relational binding**
>
> **A8.** Thank you for this important point. In our work, we also employ token-level ablations (i.e., masking or replacing individual input tokens to observe changes in the model’s output). Please also kindly refer **A7** for relational binding. We will include these related works in the revised version.

---

### Official Review · Reviewer_VaRV · 2025-03-10

**Overall Recommendation:** 3

**Summary:**

This paper studies how semantic dependency changes within the model architecture by investgating the toekn values. Through experientment, the author find that: 1) most token retain original information as layer goes deeper. 2) truthful semantic information is encoded in the token in final layer. 3) wrong output is related to incorrect semantic dependencies. And the author finds that wrong and correct semantic information is encoded in same parameter, which causes difficulties to remove incorrect semantics.

## update after rebuttal

After rebuttal I opt to update the score to 3 since the responses have addressed my most concerns.

**Claims And Evidence:**

chapter 3

I think the experiment is not convincing enough. Because of resnet,  it's reasonable and apparent that the i-th token is the most sensitive one in final layer to change of the i-th token, which is not enough to draw the conclusion(most token retain their original semantic information).

chapter 5.1

The conclution that "higher percentages suggest that their architecture may be more susceptible to false dependencies when mistakes occur" sholud be supported by more evidence. I think the percentage that model makes mistakes when maximum dependency score for incorrect answers exceeds that of correct answers,should also be calculated.

The conclusion that "lower percentages, suggesting a potentially more robust mechanism for reducing the influence of false dependencies on outputs" is wrong. Lower percentage means false dependency accounts for a small proportion in failed QA instances, which means there may be some other factors lead to wrong outputs. The conclusion "reducing the influence of false dependencies on outputs" is unreasonable.

To make the analysis and reasult in 5.1 more convincing, it's helpful to provide an table similar to the confusion matrix.

**Essential References Not Discussed:**

The key contribution of the paper is analyzing how semantic dependencies affect token behavior in transformer models, but it does not cite recent work on semantic role labeling (SRL) advancements, such as https://arxiv.org/abs/2502.08660, which introduces a novel method for capturing fine-grained semantic dependencies in transformers. This work could provide additional context for understanding how semantic dependencies are encoded and propagated across layers.

**Experimental Designs Or Analyses:**

Discussed in *Claims and evidence, Methods and Evaluation Criteria.* section.

**Methods And Evaluation Criteria:**

The paper mainly uses perturbation method to calculate semantic dependency between token in first layer and in final layer. I think the selected perturbation token should not be semantically similar to the original token. more detailed explanation to token perturbation (like the vocabulary) should be provided.

 The author compares different large models(Bert and GPT) to demonstrate the findings, but more recent open source models could also be considered. And it will be more objective to list parameter size in the table.

**Other Comments Or Suggestions:**

A typo:

In equation(3), $FFN(z^l)$ should be $FFN(\hat{z}^l)$

**Other Strengths And Weaknesses:**

Discussed above.

**Questions For Authors:**

According to figure1 and validation2 in table1, is the causal mask removed in GPT-2 and LLaMA3?

**Relation To Broader Scientific Literature:**

The paper could strengthen its contribution by explicitly linking its findings to existing research on semantic dependencies and transformer models, highlighting how it advances or diverges from prior work.

**Theoretical Claims:**

no theoretical claim and proof

---

> ### Author Rebuttal · Authors · 2025-04-01
>
> **Q1. (Claims, Chapter 3) Because of resnet, it's reasonable that the i-th token is the most sensitive one in final layer to change of the i-th token, which is not enough to draw the conclusion that most token retain their original semantic information.**
>
> **A1.** Thank you for this insightful observation regarding the influence of residual connections. We are not sure if we have interpreted your comment correctly, please kindly allow us to elaborate on our main point.
>
> - Our central claim is that semantic information encoded in a final-layer token can be observed by measuring its sensitivity to changes in the corresponding input token. This claim is independent of any specific architectural feature, such as residual connections.
> - Your observation is highly insightful. Residual shortcuts could encourage final-layer tokens to retain their original semantic information. For example, in a very simple 1-layer model with a direct shortcut from input to output, it is highly likely that the final-layer token closely reflects the original input token.
>
> Additionally, we conducted additional experiments on different GPT model sizes.
>
> - Interestingly, although all models share the same residual architecture, larger models (e.g., GPT-2-XL) exhibit significantly stronger semantic retention at the final-layer token level than smaller models (e.g., GPT-2).
> - This is somewhat counterintuitive. One might expect that simpler models, like GPT-2, would retain more of the original token content due to fewer layers and less transformation. However, the results indicate oppsitely.
> - This suggests that semantic retention is also influenced by other factors such as model complexity.
>
>
> ---
>
> **Q2. (Claims, Chapter 5.1) To make the analysis and result in 5.1 more convincing, it's helpful to provide a table similar to the confusion matrix.**
>
> **A2.**  Thank you for the insightful comment! To further make the analysis in Section 5.1 more comprehensive, we have followed your comment and included a table for each model. Due to space limitations, the tables will be attached later.
>
> The results show that when the model correctly encodes the semantic dependency in the final-layer token, it usually provides the correct answer. Conversely, when the model produces an incorrect answer, the semantic dependency is often incorrectly encoded.
>
> These findings highlight the importance of semantic dependency encoded in the final-layer token for model predictions.
>
> ---
>
> **Q3. (Claims, Chapter 5.1) I think the percentage that the model makes mistakes when the maximum dependency score for incorrect answers exceeds that of correct answers, should also be calculated.**
>
> **A3.** The metric in our original paper explicitly follows your method. We have revised our paper by including your comments to make this point clearer.
>
> ---
>
> **Q4. (Claims, chapter 5.1) The conclusion "reducing the influence of false dependencies on outputs" is unreasonable.**
>
> **A4.** We are sorry for this terrible confusion.
>
> We intended to say that RoBERTa shows a lower proportion (69.20%) compared to TinyRoBERTa (77.94%), but a similar L1 score. This may indicate that RoBERTa has a more robust mechanism for reducing the influence of false dependencies on outputs.
>
> We found your comment “a lower percentage means false dependencies ...”  is an important point, and we have carefully acknowledged it in the revised version.
>
> Thank you again for your valuable feedback and for helping us improve the quality of the paper.
>
> ---
>
> **Q5. (Methods) I think the selected perturbation token should not be semantically similar to the original token.**
>
> **A5.** Thank you for being so insightful. We follow the same idea that the perturbation token is randomly sampled from the full vocabulary of each model. The probability of selecting a semantically similar token is low.
>
> ---
>
> **Q6. (Methods)  Recent open source models could be considered.**
>
> **A6.** The result will be provided shortly.
>
> ---
>
> **Q7. (References) Semantic role labeling (SRL) advancements should be included.**
>
> **A7.** Thank you for your suggestion. We have included and added a discussion in Appendix A.1.
>
> - Semantic role labeling methods assign semantic roles to words in a sentence, which are similar to semantic dependency parsing methods.
> - Our method aims to explain how semantic dependencies are encoded in the final-layer tokens of transformers.
> - To evaluate whether transformer models encode truthful semantic dependencies in their final-layer tokens, semantic role labeling methods can be leveraged as a reference.
>
> ---
>
>
>
> **Q8. (Questions) According to figure1 and validation2 in table1, is the causal mask removed in GPT-2 and LLaMA3?**
>
> **A8.** Thank you for your question. The causal mask is not removed. When an input layer token changes, final layer tokens in autoregressive models like GPT-2 and LLaMA3 exhibit zero change for tokens on its left (see Appendix A.3). We will emphasize this to avoid further confusion.

---

> > ### Comment · Reviewer_VaRV · 2025-04-04
> >
> > Thank you for your reply. With your revision in Chapter 5, I believe the relationship between the wrong output and incorrect semantic dependence will be more clear. However, I still think the experiment setting in Chapter 3 (Most Tokens Primarily Retain Their Original Semantic Information Through Transformer Layers) fails to provide convincing evidence for the existence of semantic dependency mechanisms in the model. Could you provide more informational experiments and evidence in charpter3（Q1）, since I think  resnet is largely responsible for the results in your experiment setting.
> >
> > Thanks for the Reply Rebuttal Comment by authors. I will update my score from negative to positive.

---

> > > ### Author Response · Authors · 2025-04-05
> > >
> > > Dear Reviewer VaRV,
> > >
> > > ### Thank you for your constructive comments, which have helped strengthen our paper. We believe this point is both important and insightful. We have followed your suggestions and the details are as follows:
> > >
> > > ---
> > >
> > > ###  1. Additional experiments
> > >
> > > ***Experimental Setup***
> > >
> > > To demonstrate that the i-th final-layer token primarily retains the semantic information of the i-th input token, we test how well it predicts the identity of input tokens at various positions. The key idea is: if the i-th final-layer token is more predictive of the i-th input token than of tokens at other positions, it indicates that it primarily retains the semantic information of the i-th input token.
> > >
> > > ***Dataset Generation (an example):***
> > >
> > > 1. **Vocabulary Selection:**
> > >     - Choose four tokens randomly from the vocabulary, e.g., T₁ = *dog*, T₂ = *eat*, T₃ = *fire*, T₄ = *place*.
> > >     - We construct 7-token sentences in which we fix the tokens at the (i‑1) and i positions while randomly sampling the other tokens from the vocabulary.
> > > 2. **Synthetic Examples:** Generate 10,000 examples by constructing sentences where the (i‑1)-th and i‑th input tokens take on the following combinations (2,500 examples each):
> > > - (i‑1, i) = (T₁, T₃)
> > > - (i‑1, i) = (T₂, T₃)
> > > - (i‑1, i) = (T₁, T₄)
> > > - (i‑1, i) = (T₂, T₄)
> > >
> > > 8000 for training and 1000 for validation and 1000 for test.
> > >
> > > ***Evaluation Procedure:***
> > >
> > > - Use a pre-trained model (e.g., BERT) to extract the i‑th final-layer token representation from each sentence.
> > > - Train two binary classifiers using these representations:
> > >     - i‑th token classifier: predicts T₃ vs. T₄ (i‑th input token identity)
> > >     - (i‑1)-th token classifier: predicts T₁ vs. T₂ ((i‑1)-th input token identity)
> > > - Evaluate both classifiers and compare their test accuracies.
> > > - Repeat the experiment 10 times and record how often the classifier for the i‑th token achieves higher accuracy than the classifier for the comparison position (e.g., i‑1).
> > >
> > > ***Generalization:***
> > > We repeat the above procedure by varying the comparison position across different offsets relative to i:
> > >
> > > i vs. i‑3, i vs. i‑2, i vs. i‑1, i vs. i+1, i vs. i+2, and i vs. i+3.
> > > In each case, we fix the input tokens at position i and a comparison position (e.g., i–1, i+1), and evaluate which token is better predicted from the i-th final-layer token. We test representative models and conduct a total of 360 experiments.
> > >
> > > ***Results:***
> > > The table below shows the percentage of trials (out of 10) in which the i‑th token identity is predicted more accurately than the comparison token identity:
> > >
> > > | Model | i vs i‑3 | i vs i‑2 | i vs i‑1 | i vs i+1 | i vs i+2 | i vs i+3 |
> > > | --- | --- | --- | --- | --- | --- | --- |
> > > | BERT | 100% | 100% | 100% | 100% | 100% | 100% |
> > > | LLama | 100% | 100% | 90% | 100% | 100% | 100% |
> > > | GPT‑2 | 100% | 90% | 80% | 100% | 100% | 100% |
> > >
> > > Results show that the i-th final-layer token is most predictive of the i-th input token and therefore primarily carries the semantic information about the i-th input token.
> > >
> > > Note that the result mentioned in A1 about resnet structure is included in the link:
> > >  https://files.catbox.moe/x1uc3n.pdf
> > >
> > > ---
> > >
> > > ### 2. further explanation of our approach in Chapter 3
> > >
> > > - We are interested in how much semantic information about the original input token is retained in its final-layer token.
> > > - This falls under the umbrella of dependence between the input token and the final-layer token. If there is high dependence, it means that the final-layer token is highly predictive of input token.
> > > - To measure dependence, we change an input token and observe which final-layer token changes the most. If a token changes significantly, it suggests high dependence.
> > > - In our experiments, to test whether the i-th final-layer token primarily retains the i-th input token’s semantic information, we perturb input tokens at different positions and observe the dependence in the i-th final-layer token.
> > > - We found that when the i-th input token is changed, the i-th final-layer token changes the most. This implies that the i-th final-layer has strongest dependence with i-th input token but not others.
> > > - The observation suggests that i-th final-layer token is the most predictive to the i-th input token but not others. Therefore, the i-th final-layer token primarily retains the semantic information of the i-th input token.
> > >
> > > ---
> > >
> > > ### As noted in our rebuttal, the results table will be attached later; we have included them in the anonymous link for reference. Below is a brief summary for your convenience:
> > >
> > > - Q2: Added two-by-two possibility tables (Table 8), making results more comprehensive.
> > > - Q6: Included additional experiments with the open-source Qwen model (Tables 9–11), showing consistent support for all claims.
> > > - We also conducted many extra experiments to further strengthen the paper, including using multiple semantic parsing methods, using GPT-4o for accurate answer evaluation, comparing GPT and LLaMA3 on a QA task under a 1-shot setting, etc.

---

### Official Review · Reviewer_LWrx · 2025-03-10

**Overall Recommendation:** 4

**Summary:**

The authors investigate a way to measure token dependency and how varying levels of dependency affect transformer model performance, contribute to incorrect information, and encode semantic dependencies. Analyzing BERT, LLaMA, and GPT, they find that most tokens retain their original semantic meaning, with final-layer tokens usually encoding truthful dependencies, though they remain sensitive to context changes and order. They further find that errors arise when certain tokens falsely encode semantic dependencies, and understanding these mechanisms is hypothesized to enhance transformer model accuracy.

## update after rebuttal
I have revised my overall score to 4 (accept) as the paper makes a relevant contribution to structured explainability and presents solid empirical results. I think the paper misses some key prior works in detecting token interactions as summarized below. Assumign these minor changes to be implemented until the camera-ready version, I would vote for accepting the paper.

**Claims And Evidence:**

- Claim 1: Most tokens primarily retain their original semantic information, even as they pass through the layers of transformers.
    - Evidence:  Table 1 presents the fraction of tokens that retain their original information.  The score defined in Eq. 6 defines retaining original information as resulting in maximal change for token j when perturbing i.
    - Reviewer Evaluation: The evaluating seems suitable to assess this claim, it remains unclear how much these results depend on the specific run of sampling random tokens. Also the choice of the L2 distance to assess semantic dependency is not clearly motivated, other distance functions may have been used, e.g. dot products, cosine similarities, etc. It was not entirely clear to me how scores of validation 2 were calculated.

- Claim 2: A token in the final layer usually encodes truthful semantic dependency.
    - Evidence: By using dependency trees the agreement between these and dependency scores is computed (Table 2).
    - Reviewer Evaluation: This appears to be a good evaluation of the claim (given the limitations raised of the dependency scores raised in Claim 1) and supports the evaluation.

- Claim 3: Model mistakes are linked to certain tokens that incorrectly encode information that is not semantically dependent.
    - Evidence: Table 3 presents the fraction of failed answers on the SQuAD Q&A dataset when assessing the dependency strength     -between the answer and question token. It creates an empirical connection between wrong answer tokens having a stronger effect on the question token than the correct answer token.
    - Reviewer Evaluation: This appears to be a simple yet sound evaluation approach.

**Essential References Not Discussed:**

- Feature attribution to assess importance of tokens (in the context of Transformers and NLP)

    - [1] Abnar, S., & Zuidema, W. (2020). “Quantifying attention flow in transformers”. ACL 2020

    - [2] Ali, A., Schnake, T., Eberle, O., Montavon, G., Müller, K. R., & Wolf, L. (2022, June). XAI for transformers: Better explanations through conservative propagation. In International conference on machine learning (pp. 435-451). PMLR.

- Semantic dependencies as interactions between features/tokens

    - [3] Eberle, O., Büttner, J., Kräutli, F., Müller, K. R., Valleriani, M., & Montavon, G. (2020). “Building and interpreting deep similarity models”, IEEE Transactions on Pattern Analysis and Machine Intelligence, 44(3), 1149-1161.

    - [4] J. D. Janizek, P. Sturmfels and S. Lee. “Explaining explanations: Axiomatic feature interactions for deep networks”, CoRR, 2020.

    - [5] Schnake, T., Eberle, O., Lederer, J., Nakajima, S., Schütt, K. T., Müller, K. R., & Montavon, G. (2021). “Higher-order explanations of graph neural networks via relevant walks”. IEEE transactions on pattern analysis and machine intelligence, 44(11), 7581-7596.


- Feature binding in language

    - [6] Vasileiou, A. and  Eberle, O.. “Explaining Text Similarity in Transformer Models”, NAACL 2024.
    - [7] Feng J. and Steinhardt, J.  “How do Language Models Bind Entities in Context?”, ICLR 2024.

**Experimental Designs Or Analyses:**

- The experimental design is exhaustive and covers a representative set of models/architectures and tasks.
- Analyses appear sound and appear reproducible with reasonable efforts.

**Methods And Evaluation Criteria:**

- The methods are clearly defined and appear to be reproducible with reasonable efforts.
- The method may suffer from some limitations (see below) that are not sufficiently discussed.
- The selection of randomly selecting tokens may introduce out of domain predictions that can result in unreliable results. Alternatives would be to measure sensitivity or relevance via gradients/feature attribution.
- The choice of the L2 distance to assess semantic dependency is not clearly motivated.

**Other Comments Or Suggestions:**

- Caption of Figure 1 should more clearly state what validation 1 and 2 are.
- Overall, I think it is a good paper that investigates clearly defined claims from an empirical perspective. The novelty is moderate and focused on language experiments.

**Other Strengths And Weaknesses:**

Strengths:
- The provided illustrations are well done and helpful to understand the approach.
- The paper is overall very well written and clearly structured.
- The discussion of the results was good and provided additional depth.


Weaknesses:
- Lack of discussing related approaches in the interpretability and explainable AI community.
- Lack of theoretical contributions to better understand what drives feature dependency, i.e. is it directly related to the magnitude of the attention score?

**Questions For Authors:**

- How strongly does the dependency at a given layer correspond to the strength of the attention score?

**Relation To Broader Scientific Literature:**

The work overall does a good job at contextualizing and motivating the results. It lacks some related works for the interpretability community that has come up with a variety of methods and approaches to investigate feature importance and feature interactions (see below), which should be added to the final manuscript.

**Theoretical Claims:**

N/A

---

> ### Author Rebuttal · Authors · 2025-04-01
>
> **Q1. (Claim 1) How much do these results depend on the specific run of sampling random tokens?**
>
> **A1.** Thank you for this important question. To ensure the stability of our results, we performed 5 independent random sampling trials for each token in the sequence and reported the average score across these trials. This ensures that our findings are not overly sensitive to a specific random sample.
>
> Following your suggestion, we conducted additional experiments using 10 independent random samples per token. The results remained very similar, further validating the stability of our results.
>
> ---
>
> **Q2.  (Claim 1 and Methods)  The choice of the L2 distance but not other distance functions, e.g. dot products, cosine similarities.**
>
> **A2.** Thank you for this insightful comment! Following your comments.
>
> We followed your comment and carefully added a discussion on our assumption for using  L2 distance in our revised version, including examples. Thank you for helping us further improve our paper.
>
> We chose L2 distance based on the assumption that both magnitude and direction of the representation vector contribute to changes in semantic meaning. In contrast:
>
> - Cosine similarity captures only directional changes, completely ignoring magnitude.
> - Dot product can reflect magnitude to some extent but is heavily dependent on direction
>
> To illustrate this, we will provide illustrative examples later due to limited space.
>
> ---
>
> **Q3.  (Claim 1)  How scores of validation 2 were calculated.**
>
> **A3.** Intuitively, we computed the Validation 2 scores by modifying one input token at a time and measuring the percentage of tokens in the final layer whose representations change. We found that nearly all final-layer tokens are affected, suggesting that information from a single input token is distributed, though to varying degrees, across the entire final layer.
>
> It’s worth emphasizing that this is not a major finding of our paper, therefore, we presented the results in the appendix. The only reason we highlight it in a few words in the main text is to draw attention to a key contrast with human language processing—humans seem do not propagate the information of a word to all others.
>
> We do not yet know whether this behavior is beneficial or harmful for model performance, but we believe it warrants attention and further investigation.
>
> ---
>
> **Q4. (Methods) Random token selection may introduce out-of-domain predictions, which leads to unreliable results.**
>
> **A4.** Thank you for this very insightful point! We have added your comment to Discussion section of our revised version.
>
>  We believe measuring sensitivity or semantic relevance through gradients could provide valueble insights and are exciting directions for future work.
>
> ---
>
> **Q5. (Essential References, Weaknesses)** The work overall does a good job at contextualizing and motivating the results. It lacks some related works from the interpretability community.
>
> **A5.** Thank you very much for your insightful comments regarding related work. Incorporating these references can significantly strengthen our paper by providing a more comprehensive overview of the interpretability literature and making the work more self-contained.
>
> Following your suggestion, we have revised the paper and carefully discussed the mentioned references in Appendix A1.
>
> Additionally, we highlight key differences between our approach and existing lines of work:
>
> - **Feature attribution methods** primarily aim to assess the importance of individual tokens or features to the model’s output.
> - **Semantic dependency methods based on feature/token interactions** focus on study the contribution of combinations of features or tokens to model predictions.
> - **Feature binding methods** often do not test whether the model’s most confident output reflects encoded semantic dependencies, rather, many assume this relationship holds and to study downstream properties. In contrast, our method is designed to explicitly test the assumption by evaluating whether there is a dependence between the model’s output and the semantic dependency encoded in the final-layer token.
>
> ---
>
> **Q6. (Weakness) Lack of theoretical contributions. Is feature dependency related to the magnitude of the attention score? How strongly does the dependency at a given layer correspond to the strength of the attention score?**
>
> **A6.** Thank you for the thoughtful comment.
>
> Yes, we believe that feature dependency is directly related to the magnitude of the attention score. A mathematical formulation is provided in Appendix A.4.
>
> However, deriving a precise theoretical relationship under realistic assumptions between attention score and semantic dependency is very challenging due to the nonlinear and complex structure of neural networks. We acknowledge this limitation in our revised version and recognize it as an important direction for future research.

---

### Official Review · Reviewer_gFV2 · 2025-03-13

**Overall Recommendation:** 4

**Summary:**

The manuscript explores how transformer-based language models encode semantic dependencies and how semantic dependencies contribute to errors in model outputs. The authors propose a perturbation-based interpretability method to measure semantic dependencies. They mainly examined how changes in input tokens affect token representations within the models. There are mainly four discoveries:

1. Tokens mostly retain original semantic information through transformer layers.

2. Models generally encode truthful semantic dependencies in their final layers.

3. Model mistakes frequently arise due to falsely encoded semantic dependencies

4. Correcting these mistakes via direct parameter adjustments is challenging because the same parameters encode both correct and incorrect semantic dependencies.

They validate these findings with extensive experiments across various transformer architectures including BERT variants, GPT-2, and LLaMA 3, primarily using perturbation-based techniques on texts of diverse sources such as GSM8K, Yelp, and SQuAD.

**Claims And Evidence:**

The key claims in the paper are supported by extensive experiments, although some limitations exist.

## **Claim 1**
Tokens mostly retain original semantic information through transformer layers. This claim is supported by high retention percentages in Table 1, though a deeper analysis and discussion on the differences between GPT-2’s 75% vs. BERT’s 98.8% would better support this claim. Overall, the claim is supported by the empirical evidence.

## **Claim 2**
Models generally encode truthful semantic dependencies in their final layers. High alignment scores in Table 2 validate this claim. One thing that might need attention is that SpaCy labels instead of human expert annotations are used as ground truth, which might incur bias inherent to SpaCy's models. Alternatively, I would suggest either including multiple automatic semantic dependency parsing, or manually annotate a small subset of the dataset to verify the consistency and strengthen the conclusion. Also, the case when a word spans multiple subword tokens is not considered, which might need some brief additional justification given that is a common scenario.

## **Claim 3**
Model mistakes frequently arise due to falsely encoded semantic dependencies. The claim is supported by QA task results (Table 3). I'm a bit confused how the F1<0.6 threshold for errors is selected, and despite the generative nature, how GPT-2 resulted in an F1 as low as 0.78%, some qualitative examples would help.

## **Claim 4**
Correcting the mistakes via direct parameter adjustments is challenging because the same parameters encode both correct and incorrect semantic dependencies. Visualization (Figure 5 and Appendix A.4) shows that certain attention heads encode both correct and incorrect dependencies, and directly disabling these heads, such as head pruning, might hurt overall performance. However, I believe a word other than "adjustment" should be used to make the claim more accurate

**Essential References Not Discussed:**

Not that I am aware of. The detailed related works in A.1 are rather comprehensive.

**Experimental Designs Or Analyses:**

Experiment designs are overall sound and empirical evidence is rather thorough. Optionally I would encourage:
* Validating SpaCy-derived dependencies against human annotations, or cross validate with other tools.
* Testing perturbation with synonym tokens (vs. random).

**Methods And Evaluation Criteria:**

## **Methods**
The authors mainly used the perturbation-based method to trace the semantic flow in transformers. One minor suggestion is that random vocabulary replacement might introduce noise as it might drastically change the sentence into nonsensical texts. I'm wondering whether using controlled perturbations (e.g., synonyms) instead of random vocabulary could help strengthen validity.

## **Evaluation Criteria**
The authors tested their claims against a battery of datasets, which spans multiple domains (math, reviews, wiki, etc.) and tasks (QA, classification, etc.). The datasets used in this paper are adequate for the authors' purpose and are representative of broader research interest. The evaluation metrics used (both newly defined and existing) are reasonable and make convincing support for the authors' claims.

**Other Comments Or Suggestions:**

Typos:

Figure 4b: marry -> Mary; samantic -> semantic

L248: Spacy -> SpaCy

**Other Strengths And Weaknesses:**

## **Strengths**
* The paper is well written and easy to follow. The details are clearly explained.
* The core method is novel, simple and effective, and can be helpful for other related probing use cases.

**Questions For Authors:**

**Q1:** Why was F1<0.6 chosen to define incorrect QA answers? How would results change with stricter/lenient thresholds?

**Q2:** In your discussion section you mentioned "For instance, replacing a token with a semantically similar yet different token may lead to significant variation depending on the model’s interpretation", can you elaborate why this will be an issue and why random sampling from the vocabulary is better?

**Relation To Broader Scientific Literature:**

Connects well with semantic dependency parsing, probing studies, and interpretability of attention mechanisms. Differentiates by focusing on token-level error analysis.

**Theoretical Claims:**

No formal theoretical proofs, but the math is overall sound. The link between parameter localization and dependency encoding could be more rigorously established as it is not very straightforward.

---

> ### Author Rebuttal · Authors · 2025-04-01
>
> **Q1. (Claim 1) Analysis of differences between GPT-2 (75%) and BERT (98.8%).**
>
> **A1.**  Thank you for your insightful finding about GPT-2.
>
> We believe that this is likely related to model complexity. Following your findings, we conducted additional experiments and calculated the percentages for GPT-2, GPT-2-Large, and GPT-2-XL across different datasets. The results show that larger GPT models achieve similar percentages to BERT, e.g., around 98%. Due to the character limit of the rebuttal, the table will be included later.
>
> ---
>
> **Q2. (Claim 2) Including multiple automatic semantic dependency parsing methods**
>
> **A2.** Thank you for the great suggestions. We have followed your comments and included results using different semantic dependency parsing methods. Specifically, we used two popular dependency parsing methods: Stanza (Stanford NLP) and AllenNLP. The results for verifying truthful semantic dependencies encoded in the final layers are similar to those obtained with SpaCy. The table will be included later.
>
> ---
>
> **Q3. (Claim 2) Cases where words span multiple subword tokens are not considered, requiring justification.**
>
> **A3.** Thanks for the comment.   Although model-estimated semantic dependencies can be easily obtained, the main challenge is that existing semantic dependency parser methods usually cannot measure dependencies at the subword level. This makes direct comparison difficult. To address this issue, we may need to manually annotate the semantic dependencies and compare them with those estimated by the models, which is costly and hard to scale. We have carefully acknowledged this in our main paper.
>
> ---
>
> **Q4. (Claim 3) How the F1<0.6 threshold for errors is selected.**
>
> **A4.** The threshold of F1 < 0.6 for identifying incorrect answers was determined empirically. Since our goal is to assess whether incorrect answers are associated with incorrect semantic dependencies, an F1 score below 0.6 indicates that over 60% of the tokens predicted by the model differ from those in the original answer—strongly suggesting the answer is likely incorrect.
>
> To further strengthen this analysis, and following existing work, we conducted additional experiments using more advanced ChatGPT models to compare the model’s answer with the ground truth and find incorrect cases. The results are similar with using F1<0.6.
>
> ---
>
> **Q5. (Claim 3) how GPT-2 resulted in an F1 as low as 0.78%, some qualitative examples would help.**
>
> **A5.** Note that to ensure a fair comparison, we evaluated LLaMA and GPT models using the same zero-shot (0-shot) setting as BERT. This is the reason that they have a low accuracy. We conducted extra experiments using a one-shot setting, which aligns with official benchmark evaluations.
>
> Below are two randomly selected qualitative examples of GPT2’s performance on QA tasks.
>
> - Question: Who was the Super Bowl 50 MVP?
> - Context: The Broncos took an early lead in Super Bowl 50 and never trailed. Newton was limited by Denver's defense, which sacked him seven times and forced him into three turnovers, including a fumble which they recovered for a touchdown. Denver linebacker Von Miller was named Super Bowl MVP, recording five solo tackles, 2½ sacks, and two forced fumbles.
> - Expected Answer: Von Miller
> - GPT-2 Answer: Peyton Manning
> - Analysis: Peyton Manning is not in the context. The model incorrectly associates Super Bowl 50 MVP with Peyton Manning instead of Von Miller.
> - Question: How many times was Cam Newton sacked?
> - Context: (Same as above)
> - Expected Answer: Seven
> - GPT-2 Answer: Cam was sacked three times
> - Analysis: The model misinterprets the numerical information.
>
> ---
>
> **Q6. (Q2 in review) I'm wondering whether using controlled perturbations (e.g., synonyms) instead of random vocabulary could help strengthen validity.**
>
> **A6.** Thank you for this insightful question. To make it clearer, we have added more justification for using random vocabulary in our revised version. For both claim 2 and claim 3, we need to understand how semantic dependency is encoded in the final layer by replacing an input token *A* and observing which final-layer token changes (e.g., at position *B*).
>
> - Suppose that *A* and *B* have a strong dependence, that is, the semantic information of *A* is encoded in the final-layer representation at position *B*.
> - What we want to observe is that when *A* is replaced with another token, *B* changes significantly. This would suggest that *B* encodes information from *A*, indicating a semantic dependency between them.
> - However, if we replace *A* with a synonym (*A’*), the overall semantic meaning of the sentence may remain largely unchanged, and the model may treat *A* and *A’* similarly.
> - In this case, we may not observe a large change at *B*, making it difficult to conclude whether *B* was originally dependent on *A*, even if a true dependency existed. Therefore, we use random tokens to encourage semantic independence.

---

### Decision · Program_Chairs · 2025-05-01

**Decision:**

Accept (poster)

**Comment:**

This paper examines how transformer models represent semantic dependencies in question answering. It presents four significant observations from the study, which most reviewers find valuable to the community.
The paper can be further improved by providing a more comprehensive overview of related works from the community. It would be great if the authors can discuss how the findings in this paper can benefit other practitioners in the community.